# Targeted anticancer pre-vinylsulfone covalent inhibitors of carbonic anhydrase IX

Aivaras Vaškevičius[1], Denis Baronas[1], Janis Leitans[2], Agnė Kvietkauskaitė[1], Audronė Rukšėnaitė[3], Elena Manakova[4], Zigmantas Toleikis[5], Algirdas Kaupinis[6], Andris Kazaks[2], Marius Gedgaudas[1], Aurelija Mickevičiūtė[1], Vaida Juozapaitienė[1], Helgi B Schiöth[7], Kristaps Jaudzems[8], Mindaugas Valius[6], Kaspars Tars[2], Saulius Gražulis[9], Franz-Josef Meyer-Almes[10], Jurgita Matulienė[1], Asta Zubrienė[1], Virginija Dudutienė[1], Daumantas Matulis[1]*

[1]Department of Biothermodynamics and Drug Design, Institute of Biotechnology, Life Sciences Center, Vilnius University, Vilnius, Lithuania; [2]Latvian Biomedical Research and Study Centre, Riga, Latvia; [3]Department of Biological DNA Modification, Institute of Biotechnology, Life Sciences Center, Vilnius University, Vilnius, Lithuania; [4]Department of Protein - DNA Interactions, Institute of Biotechnology, Life Sciences Center, Vilnius University, Vilnius, Lithuania; [5]Sector of Biocatalysis, Institute of Biotechnology, Life Sciences Center, Vilnius University, Vilnius, Lithuania; [6]Proteomics Center, Institute of Biochemistry, Life Sciences Center, Vilnius University, Vilnius, Lithuania; [7]Functional Pharmacology and Neuroscience, Department of Surgical Sciences, Uppsala University, Uppsala, Sweden; [8]Latvian Institute of Organic Synthesis, Riga, Latvia; [9]Sector of Crystallography and Chemical Informatics, Institute of Biotechnology, Life Sciences Center, Vilnius University, Vilnius, Lithuania; [10]Department of Chemical Engineering and Biotechnology, University of Applied Sciences Darmstadt, Darmstadt, Germany

*For correspondence:
matulis@ibt.lt

## eLife Assessment

This paper reports the synthesis of covalent inhibitors bearing a unique fragment as a protected covalent warhead for irreversible binding to histidine in carbonic anhydrase (CA) enzymes. These findings are **important** due to the broad utility of the approach for covalent drug discovery applications and could have long-term impacts on related covalent targeting approaches. The data **convincingly** support the main conclusions of the paper.

**Abstract** We designed novel pre-drug compounds that transform into an active form that covalently modifies particular His residue in the active site, a difficult task to achieve, and applied to carbonic anhydrase (CAIX), a transmembrane protein, highly overexpressed in hypoxic solid tumors, important for cancer cell survival and proliferation because it acidifies tumor microenvironment helping invasion and metastases processes. The designed compounds have several functionalities: (1) primary sulfonamide group recognizing carbonic anhydrases (CA), (2) high-affinity moieties specifically recognizing CAIX among all CA isozymes, and (3) forming a covalent bond with the His64 residue. Such targeted covalent compounds possess both high initial affinity and selectivity for the disease target protein followed by complete irreversible inactivation of the protein via covalent modification. Our designed prodrug candidates bearing moderately active pre-vinylsulfone

esters or weakly active carbamates optimized for mild covalent modification activity to avoid toxic non-specific modifications and selectively target CAIX. The lead inhibitors reached 2 pM affinity, the highest among known CAIX inhibitors. The strategy could be used for any disease drug target protein bearing a His residue in the vicinity of the active site.

## Introduction

Covalently binding compounds performing targeted covalent inhibition (TCI) (*Bauer, 2015*; *Gillette et al., 1974*; *De Vita, 2021*; *Singh, 2022*) bear special functional groups called 'warheads', usually Michael acceptors. Initially, these compounds bind to the target protein reversibly via their specific structural features that recognize the target protein (*Lonsdale and Ward, 2018*). In a subsequent step, an irreversible covalent bond is formed between the 'warhead' fragment and the targeted amino acid, mostly a nucleophilic one like cysteine (*Lonsdale and Ward, 2018*). This irreversible mode of binding provides a prolonged mechanism of action, full and reversible or irreversibletarget inactivation, the need for lower drug dosages, the opportunity for higher selectivity toward the target, and in some cases – effective inhibition of drug-resistant enzyme mutants (*Lonsdale and Ward, 2018*; *Singh et al., 2012*; *Sutanto et al., 2020*). Among successfully applied targeted covalent inhibition examples are ibrutinib (BTK inhibitor), afatinib (EFGR T790M mutant inhibitor), osimertinib (improved EFGR T790M mutant inhibitor), and sotorasib (KRAS G12C mutant inhibitor), which are already approved by the Food and Drug Administration (*Boike et al., 2022*; *Baillie, 2016*).

Here, we introduce pre-vinylsulfone warhead for TCI by the formation of a covalent bond with the protein histidine residue. In contrast to cysteine, the intrinsic nucleophilicity of histidine is weaker and there are few reports about histidine labeling in proteins by small molecules, most of them being highly reactive and therefore not usable as warhead in drug development due to non-specific reactions (*Jia et al., 2019*; *McCowen et al., 1951*; *Harlow and Switzer, 1990*; *Gilbert et al., 2023*). Known histidine-targeting compounds mostly carry a highly reactive warhead, such as sulfonylfluorides, which provide some degree of selectivity due to suitable substituents, which interact selectively with their target protein (*Che and Jones, 2022*; *Cruite et al., 2022*), but it is almost impossible to prevent non-specific labeling. Chemoselective modification of histidine is very difficult to achieve. A light-promoted and radical-mediated selective C-H-alkylation of histidine for peptide synthesis has been suggested (*Chen et al., 2019*), but does not apply to proteins. Another method for chemoselective histidine bioconjugation uses thiophosphorodichloridate reagents, which mimic naturally occurring histidine phosphorylation (*Jia et al., 2019*). A light-driven selective approach for labeling histidine residues in native biological systems was developed with thioacetal as thionium precursor (*Wan et al., 2022*). However, the light-driven approach requires the presence of high concentrations of Rose Bengal as a catalyst, not suitable for therapeutic applications.

We applied the TCI strategy to human carbonic anhydrases (CA), metalloenzymes that catalyze reversible $CO_2$ hydration by producing acid proton and bicarbonate anion. There are 12 catalytically active CA isozymes in humans. The CAI, II, III, VII, and XIII are cytosolic, CAIV is membrane-bound, while CAVA and VB are found in mitochondria. CAVI is the only secreted isozyme found in saliva and milk, whereas CAIX, XII, and XIV are transmembrane proteins bearing extracellular catalytic domains (*Aggarwal et al., 2013*). Catalytic domains of CA isozymes are highly homologous and bear structurally very similar beta-fold. However, the isozymes differ in their enzymatic activity, tissue distribution, and cellular localization.

CAIX is a hypoxia-inducible protein, which participates in cancer cell proliferation and metastasis (*Saarnio et al., 1998*; *Rafajová et al., 2004*). As recently demonstrated by proteomic analysis, CAIX interacts with amino acid and bicarbonate transporters to control cancer cell adhesion, a critical process involved in migration and invasion. The CAIX also plays an important role in the migration of cancer cells by interaction with collagen-, laminin-binding integrins, and MMP-14 (*Swayampakula et al., 2017*; *McDonald et al., 2018*). It is proposed that targeting CAIX catalytic activity and/or interrupting the interactions with metabolic transport proteins and cell adhesion/migration/invasion proteins will have therapeutic benefits by involving pH regulation, metabolism, invasion, and metastasis (*McDonald et al., 2018*; *Becker and Deitmer, 2021*).

Sulfonamides stand out as the most extensively researched class of CA inhibitors (*Krishnamurthy et al., 2008*; *Linkuvienė et al., 2018*). They exhibit a high binding affinity in their deprotonated state

to the zinc-bound water form of CA (*Kovalevsky et al., 2018*; *Aggarwal et al., 2016*; *Taylor et al., 1970*). Drugs used in the clinic as CA inhibitors, such as acetazolamide, methazolamide, dichloro-phenamide, dorzolamide, and brinzolamide, have various side effects (*Swenson, 2014*). The development of isozyme-selective CA inhibitors is a major goal of drug discovery. Any such drugs will be more beneficial than the currently available mostly non-selective CA inhibitors, as the reduction of side effects will improve the effectiveness of the therapy.

Covalent inhibitors have been previously designed for CAI and CAII, namely, bromoacetazolamide and N-bromoacetylacetazolamide (*Kandel et al., 1968*; *Kandel et al., 1970*; *Cybulsky et al., 1973*). Although N-bromoacetylacetazolamide formed a covalent bond with CAI His67 and bromoacetazol-amide with CAII His64 amino acids, further research of these compounds was discontinued (*Cybulsky et al., 1973*). Recently, studies involving covalent modification of CA isozymes (mainly CAII) have been published. However, the synthesized molecules were designed not to act as enzyme inhibitors, but as a model protein to investigate benzenesulfonamide-bearing fluorescent label and a warhead able to bind the enzyme covalently (*Swenson, 2014*). One of their new probes bearing an epoxide reactive group was not only able to form a covalent bond with the protein, but it did it selectively for His64 (*Chen et al., 2003*). Similarly, analogous sulfonamides without fluorescent groups were synthesized and formed the covalent bonds with His3 or His4 of CAII (*Takaoka et al., 2006*; *Wakabayashi et al., 2008*). Different warheads were applied to react with His64 and His3 (*Gilbert et al., 2023*; *Aatkar et al., 2023*) bearing S(IV) fluoride to present a new way for the expansion of the liganded proteome (*Gilbert et al., 2023*; *Aatkar et al., 2023*).

In this work, we investigated fluorinated benzenesulfonamide compounds bearing sulfonylethyl ester and sulfonylethyl carbamate moieties as possible covalent CA inhibitors. The 3-substituted-2-((2,5,6-tetrafluoro-4-sulfamoylphenyl)sulfonyl)ethyl acetate exhibited a surprisingly high binding affinity for CAIX, which was more than tenfold higher than our previous synthesized lead compound VD11-4-2 ($K_{d(CAIX)}$=32 pM) (*Dudutienė et al., 2014*). The MS and X-ray crystallography data confirmed the covalent binding of new compounds to the proton shuttle His64 residue. We showed that sulfonyl-ethyl ester/carbamate behaves as a prodrug by reacting with His64 in the active site of CAs through the elimination mechanism to release ester or carbamate moieties, thus forming a reactive vinylsul-fone group. The newly discovered mechanism of inhibition of CA through forming a covalent bond between the compound and the protein has great potential for developing high-affinity compounds for a particular CA isozyme, and such compounds could become precursors of new-generation drugs.

## Results
### Design and mechanism of covalently-binding CA inhibitors

In search of high-affinity and high-selectivity inhibitors of CAs, a series of fluoro-benzenesulfonamide-based compounds were synthesized (*Figure 1*). The benzenesulfonamide group is important for making a coordination bond with the Zn(II). At the same time, the fluorine atoms were included to withdraw electrons from the sulfonamide group and diminish its p$K_a$ to strengthen interaction. Surprisingly, part of the compounds exhibited extremely strong binding affinity. These compounds contained the -$SO_2CH_2CH_2OCOR$ group that was forming an unexpected covalent bond with the protein.

It has been known in organic chemistry that compounds bearing this fragment in the presence of bases can rearrange to vinyl-sulfonyl moiety which has been reported as a covalently modifying 'warhead' (*Tsutsui et al., 1987*; *Alonso et al., 2003*; *Ichikawa et al., 1989*). To the best of our knowledge, this kind of rearrangement/elimination has not been applied to enzyme covalent inhibitors. We designed benzenesulfonamides with the $SO_2CH_2CH_2OCOR$ group that can form highly reactive electrophilic species without adding additional base by adding multiple fluorine atoms to the benzene ring. Furthermore, the rearrangement occurred only in the CA enzyme active site. The active vinyl-sulfonyl group formed via elimination reaction, which occurred easily due to the strong electron-withdrawing effect of fluorines on the benzene ring (*Figure 2*).

The $SO_2$ group at the *para* position relative to the sulfonamide group was also necessary for a covalent bond because compounds **4** and **5** bearing the SO or S groups, respectively, did not form the covalent bond with CA isozymes. Moreover, the change of the -O-CO-R group to -NH-CO-R or -CO-O-R also prevented the formation of the covalent bond as illustrated by compounds **2**, **10**, **7**, and **8** that did not form covalent bonds with the protein molecule.

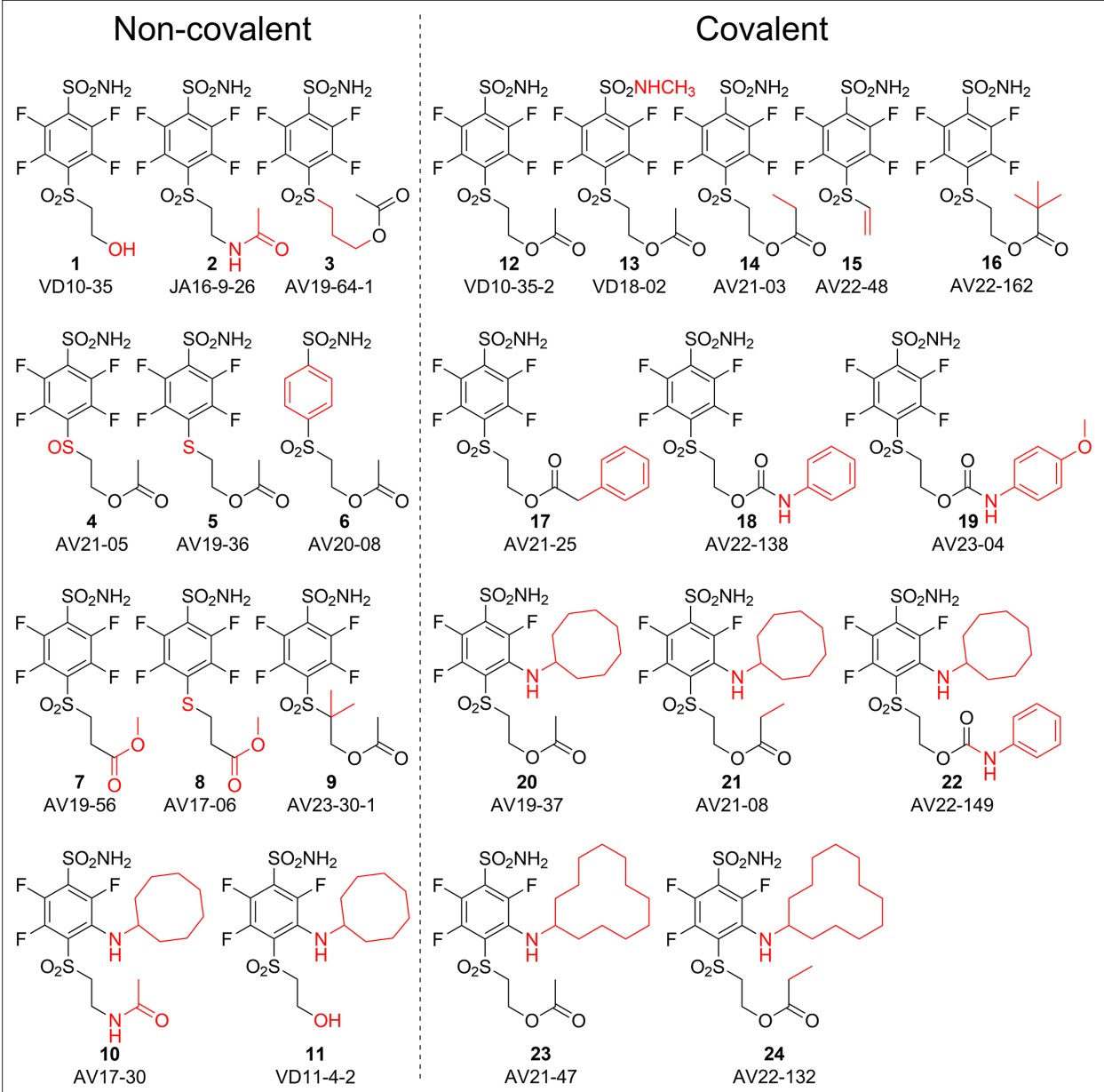

**Figure 1.** Chemical structures of compounds used in this study and designed to investigate the covalent binding capability to CA proteins. Compounds on the left of the vertical dashed line do not form the covalent bond, while the ones on the right form the covalent bond with the protein molecule. Moieties shown in red are important for structural comparison to visualize the chemical groups that are responsible for covalent interaction, high affinity, or high selectivity for CAIX.

We propose the mechanism where the covalent modification occurs via the elimination mechanism shown in *Figure 2*, where a basic amino acid residue removes the proton and forms the vinylsulfone moiety, which only then forms a covalent bond with the nitrogen atom of the histidine residue. To check this mechanism, two control compounds, **3** and **9**, were synthesized. Both of these compounds did not form a covalent bond with the protein. In compound **9**, two methyl groups located at the crucial α carbon replaced the proton that needed to be removed. In compound **3,** the third methylene group prevented the β-elimination reaction. Furthermore, the concept of formation of vinylsulfone fragment was demonstrated by synthesizing compound **15** bearing the vinylsulfone itself. This compound readily formed the covalent bond with proteins.

Since it appeared that compounds bearing ester groups (e.g. compounds **17** and **20**) may have too high rate of covalent bond formation leading to non-desired modification of non-targeted proteins, we

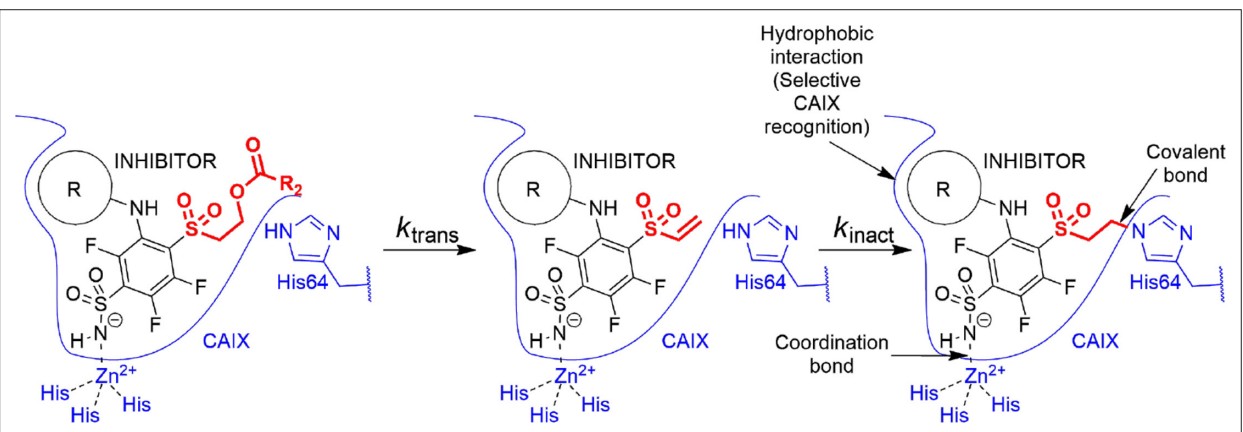

**Figure 2.** Proposed rearrangement mechanism (beta-elimination) of compounds bearing the -SO₂CH₂CH₂OCOR fragment to vinyl-sulfone and the formation of a covalent bond with the histidine (His64) amino acid side chain of the CA protein.

have designed and synthesized compounds bearing the carbamate functional group (e.g. compounds **18** and **22**). The carbamate group was more stable than the ester and thus the covalent modification reaction of the protein showed milder rate than the ester.

The cyclooctyl or cyclododecyl rings present in covalent compounds **20–24** and in non-covalent compounds **10–11** have been previously designed by our group to specifically bind to CAIX isozyme and intended not to bind to the other eleven human catalytically active CA isozymes. The ring could fit to the pocket in CAIX, but not so readily to other isozymes (*Dudutienė et al., 2014*; *Zubrienė et al., 2017*). The three-way recognition model is shown in *Figure 3*.

## Covalent interaction between inhibitors and CA isozymes by X-ray crystallography

The covalent bond between the compounds and protein was demonstrated by X-ray crystallography (*Figure 4*). The crystal structures of compounds **21**, **20**, and **23** with CAI, CAII, and CAIX, respectively, showed covalent bond formation between the ligands and the histidine residue of the enzymes. In CAI, the ligand forms a covalent bond with His 67, while in CAII and CAIX, the bond is made with His 64 – a residue responsible for proton shuttle function in CA isoforms (*Fisher et al., 2005*). The distance between the nitrogen atom of histidine and the vinyl carbon atom of the compounds was around 1.5 Å, consistent with the length of the covalent bond, and was visible with strong electron density in all three crystal structures (*Figure 4A, B and C*). The X-ray crystal structure refinement statistics are provided in *Table 1*.

The tail moiety of the ligands is also coordinated by two hydrogen bonds – with Asn 62 and Gln 92 (CAII numbering) in the cases of CAII and CAIX, while in CAI, a possible hydrogen bond is formed with Gln 92 and His 64. Electron density was overall good for the whole ligand in all three crystal structures, with slightly weaker electron density observed in the hydrophobic tail region indicating higher flexibility. The hydrophobic tail part is oriented towards the active site region, which varies

**Figure 3.** Transformation of pre-drug to the active vinyl sulfone and a three-way recognition of CA isozymes. First, the negatively charged sulfonamide forms a coordination bond with the Zn(II). Second, the hydrophobic cyclooctyl ring fits into the hydrophobic pocket of the CAIX isozyme and provides substantial selectivity over other CA isozymes. Third, a covalent bond forms with the histidine providing irreversible inhibition of CAIX enzymatic activity.

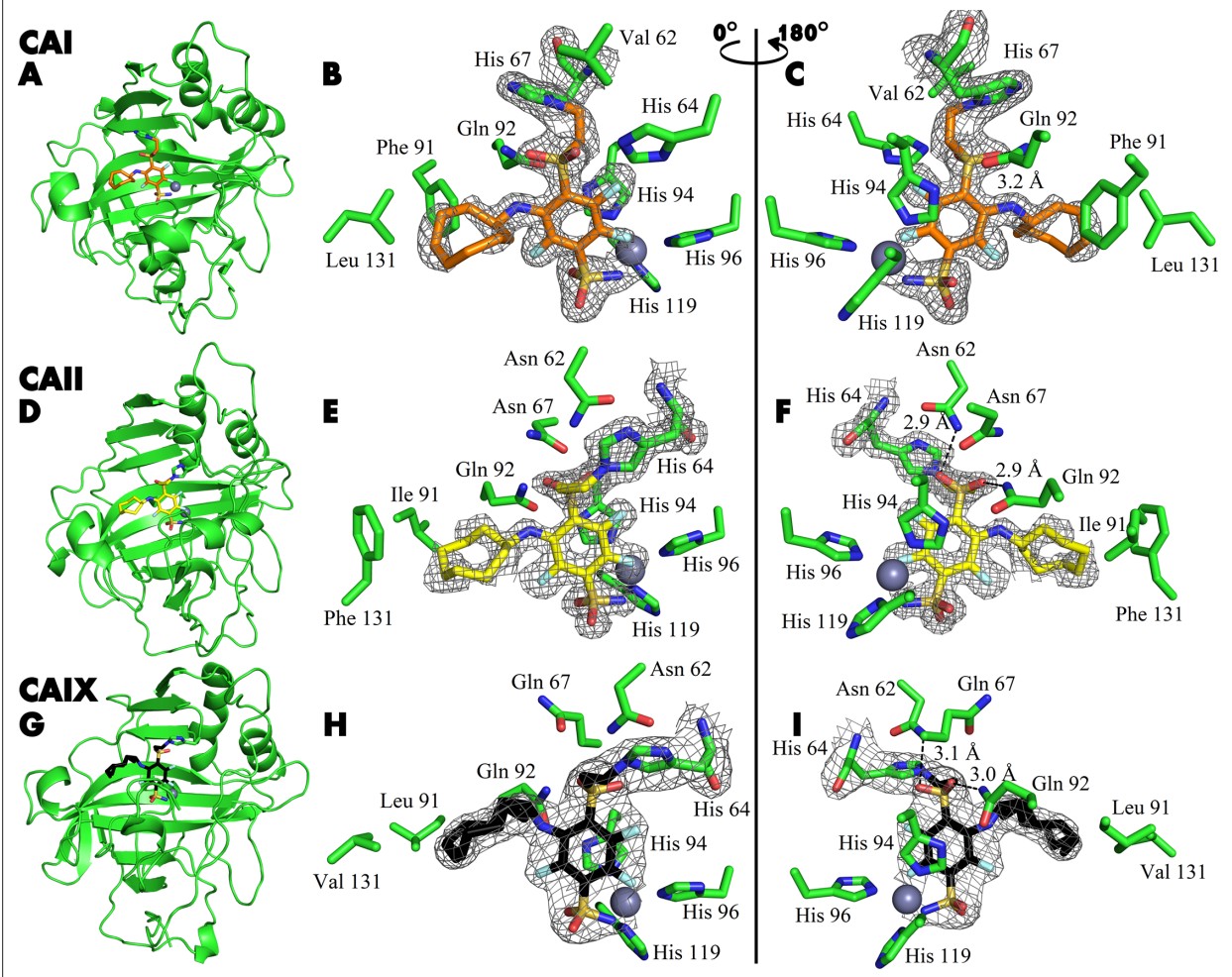

**Figure 4.** X-ray crystal structures of CAI (A–C), CAII (D–F), and CAIX (G–I) covalently bound with inhibitors 21, 20, and 23. X-ray crystal structures of CAI (A–C), CAII (D–F), and CAIX (G–I) covalently bound with inhibitors 21, 20, and 23, respectively. The left panels show cartoon models of the entire protein molecule with the covalently bound compound and the His64 residue shown as a stick model, while the middle and right panels show close-up views of the inhibitor, shown as sticks, displayed with a 180° rotation between the images. The 2Fo-Fc map is shown only for the ligand and the histidine residue with which it forms a covalent bond, contoured at 1σ.

most between CA isozymes (the so-called 'hot spot' for isozyme (isoform)-selective inhibitor design *Alterio et al., 2009*).

There are two protein subunits in the CAI structure, the position of the modeled **21** is very similar in both. The electron density of the ligand is better in protein chain B. Sulfonamide group and modification of His67 with para-linker are visible. The fluorine atoms of the benzene ring also could be located clearly, but the electron density of the benzene ring is partially lost. This could be explained by partial occupancy of the modeled conformation of the inhibitor and by rotation of ligand fraction since the ligand is probably not fixed by the covalent bond with His67. The cyclooctyl group is visible in both subunits only partially due to the flexibility of the ring resulting in multiple conformations. This group is oriented towards the hydrophobic part of the active site, defined by Leu131, Ala135, Leu141, and Leu198.

## Covalent interaction by mass spectrometry and enzymatic activity

The mass spectra of CA isoforms incubated with compound **12** showed a 319 Da shift compared to pure CA isoforms (except CAIII), equal to compound **12** molar mass without the ester group (*Table 2*). Moreover, after a 3-min incubation of 1:1 molar ratio of compound **12** with CAXIII protein (29,575 Da), nearly half of the protein was already covalently modified in this time as shown by an additional

**Table 1.** X-ray crystal structure refinement statistics of CA II-**20**, CA I-**21,** and CA IX-**23** complexes.

| Structure | CA II – 20 | CA I – 21 | CA IX – 23 |
|---|---|---|---|
| Space group | P21 | P 21 21 21 | H3 |
| Cell dimensions | | | |
| $a$ (Å) | 42.45 | 62.42 | 152.11 |
| $b$ (Å) | 41.57 | 73.27 | 152.11 |
| $c$ (Å) | 72.79 | 120.57 | 172.51 |
| $\beta$ (°) | 104.4 | 90 | 90 |
| Resolution (Å) | 70.50–1.40 | 120.57–1.39 | 52.35–2.20 |
| Highest resolution shell (Å) | 1.40–1.47 | 1.39–1.41 | 2.20–2.25 |
| No. of reflections (unique) | 47352 | 110929 | 75528 |
| No. of reflections in the test set | 2410 | 10962 | 3874 |
| Completeness (%) | 96.9 (98.0*) | 98.3 (67.6*) | 100.0 (100.0*) |
| $R_{merge}$ | 0.14 (0.48*) | 0.06 (2.7*) | 0.07 (1.16*) |
| $\langle I/\sigma I \rangle$ | 6.5 (3.0*) | 20.4 (0.5*) | 17.2 (1.9*) |
| Average multiplicity | 5.0 (5.0*) | 13.0 (4.7*) | 10.7 (11.0*) |
| R-factor | 0.18 (0.32*) | 0.21 (0.47*) | 0.17 (0.26*) |
| $R_{free}$ | 0.22 (0.34*) | 0.23 (0.48*) | 0.20 (0.32*) |
| Average B factor (Å²) | 11.0 | 26.6 | 52.4 |
| Average B factor for inhibitor (Å²) | 18.7 | 33.1 | 73.5 |
| $\langle B \rangle$ from Wilson plot (Å²) | 8.9 | 22.1 | 49.5 |
| No. of protein atoms | 2045 | 4032 | 7413 |
| No. of inhibitor atoms | 27 | 54 | 124 |
| No. of solvent molecules | 338 | 539 | 421 |
| RMS deviations from ideal values | | | |
| Bond lengths (Å) | 0.02 | 0.01 | 0.01 |
| Bond angles (°) | 1.67 | 1.85 | 1.79 |
| Outliers in Ramachandran plot (%) | 0.39 | 0 | 0.21 |
| PDB code | 8OO8 | 8S4F | 9FLF |

*Values in parenthesis are for the high-resolution bin.

29,894 Da peak (319 Da shift, *Figure 5A*). After 2 hr of incubation, essentially entire protein fraction was covalently modified by the compound. The presence of the minor peak at 30,213 Da (638 Da shift) in *Figure 5A* indicates that there is a second modification site on CAXIII protein. *Figure 5—figure supplements 1–20* show MS spectra after incubation under listed conditions of various CA isozymes with inhibitors.

The peptide mapping by digestion of the CAXIII with thrombin detected the compound **12** covalently bound exclusively to the peptide containing His64 residue (*Figure 5—figure supplements 21 and 22*). However, in the ¹H-¹⁵N-HSQC 2D NMR spectrum, upon incubation of CAII isozyme at 1:1 molar ratio for 1 hr with covalent compound **12**, in addition to the His64 signal change, we observed a decrease of peak intensity in the N-terminal part of CAII indicating an additional minor fraction of enzyme with covalently bound **12** outside CA active site (*Figure 5—figure supplements 23 and 24*). Thus, ester compound **12** may be too reactive for fully specific inhibition. The non-covalent compound **6** was incubated with CAII at a tenfold surplus of the compound, but no covalent modification was detected (*Figure 5—figure supplement 16*).

**Table 2.** Carbonic anhydrase isozyme masses in the absence of compound and incubated with covalently-modifying compound **12**.

All CA isozymes except CAIII were covalently modified by compound 12 to a variable extent.

| Enzyme | Plasmid number | Theoretical MW | Obtained mass | Protein mass with compound 12, most intense peak m/z | Difference |
|---|---|---|---|---|---|
| CA I | pL0067 | 31204.7 | 31074.31 (w/o Met) | 31393.63 | 319.32 |
| CA II | pL0059 | 29246 | 29115.54 (w/o Met) | 29434.81 | 319.27 |
| CA III | pL0066 | 31648.9 | 31518.29 (w/o Met), 31696.34 (w/o Met and glycosylated) | 31518.29 (w/o Met), 31696.34 | - |
| CA IV | pL0307 | 30454.6 | 30320.33 (w/o Met and S-S bridge) | 30639.53 | 319.20 |
| CA VA | pL0245 | 31285.3 | 31154.77 (w/o Met) | 31474.07 | 319.30 |
| CA VB | pL0173 | 34193.6 | 34063.56 (w/o Met) | 34382.29 | 318.73 |
| CA VI | pL0339 | 35367 | 35956.25 (glycosylated) 36208.06(glycosylated) 36412.74(glycosylated) 37643.18(glycosylated) 37934.87(glycosylated) | 36275.15 36528.43 36731.48 37962.27 38254.18 | 318.90 320.37 318.74 319.09 319.31 |
| CA VII | pL0137 | 31821.7 | 31689.57 (w/o Met) | 32008.73 32327.99 | 319.03 638.42 |
| CA IX | * | 28061.7 | 28060.32 | 28379.58 | 319.26 |
| CA XII | pL0119 | 29886.3 | 29754.54 (w/o Met) | 30072.81 | 318.27 |
| CA XIII | pL0058 | 29574.3 | 29574.69 | 29893.97 | 319.28 |
| CA XIV | pL0318 | 32129.7 | 31997.12 (w/o Met) 32175.25(w/o Met and glycosylated) | 32316.37 32494.53 | 319.25 319.28 |

*CA IX mutant C174S, N346Q, prepared in yeast as described in *Leitans et al., 2015*.

Therefore, a series of compounds bearing the carbamate leaving group were designed (**18**, **19**, and **22**) to reduce chemical reactivity and reduce undesired reactions with non-intended groups. Compound **22** exhibited covalent modification of CAIX, but after 4 hr incubation, there was still a significant part of non-modified protein present. Thus the reaction was significantly slower than with the ester group. The MW of CAIX as calculated from the sequence was 28061.68 Da and matched closely the measured mass of 28060.32 (*Figure 5B* top panel and *Figure 5—figure supplement 20*) or 28060.45 (lower panel). The calculated MW of compound **22** was 563.1372 Da and was measured to be 564.1444 Da. The calculated mass of compound **22** without the carbamate leaving group was 426.0895 Da, while the measured difference in *Figure 5B* was 426.41 Da, a perfect match. Thus, compound **22** exhibited both highly specific and relatively slow modification of CAIX, with a good perspective toward drug design.

The covalent irreversible and non-covalent reversible interaction was also demonstrated by comparing the inhibition of enzymatic activity of CAIX by non-covalent compound **5** and covalent compound **20** and their possibility to be dialyzed out. Both compounds fully inhibited the enzymatic activity of CAIX at 1:1 molar ratio in the same dose-dependent manner. The resultant protein-ligand complex was then subjected to 32 hr dialysis. The CAIX complex with the non-covalent **5** regained 73% of the original enzymatic activity, while the CAIX with covalent **20** did not regain any detectable enzymatic activity (*Figure 5C and D*). This indicates that the covalent modification irreversibly inhibited the enzymatic activity of CAIX.

To determine the contribution of the primary sulfonamide group on the capability of making a covalent bond with CA, we synthesized a secondary sulfonamide **13** and compared it with the analogous covalent compound **12**. After 2 hr of incubation at 10:1 molar surplus of secondary sulfonamide **13**, the free CAII protein still dominated, indicating that only a minor fraction of the protein was

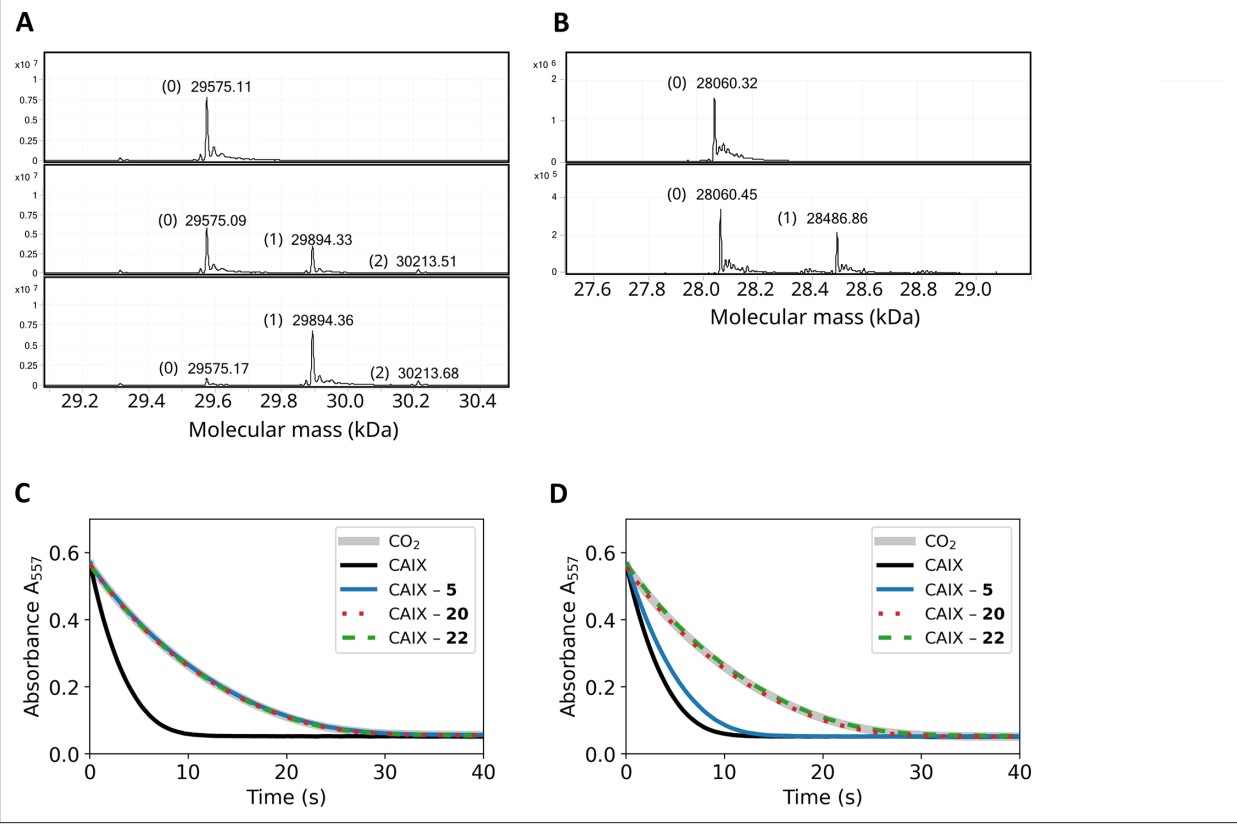

**Figure 5.** Covalent interaction shown by HRMS, and enzymatic activity recovery assay. (**A**). MS spectra of CAXIII in the absence of compound (top panel), the presence of 1:1 molar ratio of compound **12** after 3-min incubation (middle panel) and 2 hr incubation (bottom panel). (**B**). MS spectra of CAIX in the absence of compound (top panel) and after the incubation of 1:1 molar ratio of compound **22** (carbamate) for 4 hours. (**C**). Enzymatic activity of CAIX before dialysis, while **D**) – after dialysis (32 hr, 4-times buffer change, black solid line – fully active CAIX, grey solid line – spontaneous $CO_2$ hydration reaction (coincides with fully inhibited CAIX), blue solid line – CAIX with non-covalent **5**, dotted red line – CAIX with covalent **20**, and dashed green line – CAIX with covalent carbamate **22**. The recombinant CAIX recovered almost full activity after dialyzing out the non-covalent compound, while the activity remained fully inhibited with the covalent compounds.

The online version of this article includes the following figure supplement(s) for figure 5:

**Figure supplement 1.** CA I mass spectra (**A**); CA I mass spectra after incubation with **12** (**B**); CA I mass spectra after incubation with **20** (**C**).

**Figure supplement 2.** CA II mass spectra (**A**); CA II mass spectra after incubation with **12** (**B**); CA II mass spectra after incubation with **20** (**C**).

**Figure supplement 3.** CA III mass spectra (**A**); CA III mass spectra after incubation with **12** (**B**); CA III mass spectra after incubation with **20** (**C**).

**Figure supplement 4.** CA IV mass spectra (**A**); CA IV mass spectra after incubation with **12** (**B**); CA IV mass spectra after incubation with **20** (**C**).

**Figure supplement 5.** CA VA mass spectra (**A**); CA VA mass spectra after incubation with **12** (**B**); CA VA mass spectra after incubation with **20** (**C**).

**Figure supplement 6.** CA VB mass spectra (**A**); CA VB mass spectra after incubation with **12** (**B**); CA VB mass spectra after incubation with **20** (**C**).

**Figure supplement 7.** CA VI mass spectra (**A**); CA VI mass spectra after incubation with **12** (**B**).

**Figure supplement 8.** CA VII mass spectra (**A**); CA VII mass spectra after incubation with **12** (**B**); CA VII mass spectra after incubation with **20** (**C**).

**Figure supplement 9.** CA IX mass spectra (**A**); CA IX mass spectra after incubation with **12** (**B**); CA IX mass spectra after incubation with **20** (**C**).

**Figure supplement 10.** CA XII mass spectra (**A**); CA XII mass spectra after incubation with **12** (**B**); CA XII mass spectra after incubation with **20** (**C**).

**Figure supplement 11.** CA XIII mass spectra (**A**); CA XIII mass spectra after incubation with **12** (**B**); CA XIII mass spectra after incubation with **20** (**C**).

**Figure supplement 12.** CA XIV mass spectra (**A**); CA XIV mass spectra after incubation with **12** (**B**); CA XII mass spectra after incubation with **20** (**C**).

**Figure supplement 13.** CA XIII mass spectra (**A**); CA XIII mass spectra after incubation with 5 eq. of compound **3** (**B**).

**Figure supplement 14.** CA XIII mass spectra (**A**); CA XIII mass spectra after incubation with 5 eq. of compound **9** (**B**).

**Figure supplement 15.** CA XIII mass spectra (**A**); CA XIII mass spectra after incubation with 2 eq. of compound **15** (**B**).

**Figure supplement 16.** CA II mass spectra (**A**); CA II mass spectra after incubation with 10 eq. of compound **6** (**B**).

**Figure supplement 17.** CA II mass spectra (**A**); CA II mass spectra after incubation with 10 eq. of compound **13** (**B**); CA II mass spectra after incubation with 10 eq. of compound **12** (**C**).

*Figure 5 continued on next page*

*Figure 5 continued*

**Figure supplement 18.** CA XIII mass spectra (**A**); CA XIII mass spectra after incubation with 10 eq. of compound **13** (**B**).

**Figure supplement 19.** CA XIII Mass spectra (**A**); mass spectra ~3 min after addition of 1 eq. of **18** (**B**); mass spectra 40 min after addition of 1 eq. of **18** (**C**); mass spectra 2 hr after addition of 1 eq. of **18** (**D**); mass spectra 80 min after addition of 10 eq. of **18** (**E**).

**Figure supplement 20.** CA IX mass spectra (**A**); CA IX mass spectra after incubation with 2 eq. of compound **22** (**B**).

**Figure supplement 21.** Identified peptides after trypsinization of CAXIII protein in the presence of compound 12 (VD10-35-2).

**Figure supplement 22.** The peptide fragments that appear after fragmentation in the spectrometer.

**Figure supplement 23.** The change of His64 position of CAII in 1H-15N-HSQC spectrum: the control (**A**); after the addition of non-covalent sulfonamide **1** (**B**); after the addition of covalent (bottom panel) inhibitor **12** (**C**).

**Figure supplement 24.** Intensity change and chemical shift perturbation (CSP) of recombinant human carbonic anhydrase peaks in the 1H-15N-HSQC spectrum after the addition of non-covalent (orange) or covalent (black) inhibitors.

covalently modified (*Figure 5—figure supplement 17B*). In comparison, using the same conditions, compound **12** bearing the primary sulfonamide group completely modified CAII (*Figure 5—figure supplement 17C*). This shows the significant effect of the primary sulfonamide group in guiding the compound into the CA active site and consequent covalent bond formation with His64 amino acid. In the absence of the guiding sulfonamide group, such as in **13**, a relatively slow modification most likely occurred on nucleophilic residues different than the His64 in the protein active site. This unintended covalent modification by the secondary-sulfonamide **13** was also observed with isozymes other than CAII. Although this compound should have low affinity to all CA isozymes, it still modified CAXIII at a 10:1 compound surplus molar ratio after 2 hr incubation (*Figure 5—figure supplement 18*). Despite that, it is important to note that the modification most likely occurred at a different nucleophilic amino acid, not the His64 in the active site.

The presence of non-specific unintended covalent modifications prompted us to synthesize different covalently modifying groups that would be less reactive and more suitable for drug design, such as carbamate compounds **18** and **19**, which showed significantly slower (at least by two orders of magnitude) covalent-modification activity compared to the ester compounds (*Figure 5—figure supplement 19*). Even using less reactive carbamates, CA isozymes were still able to make covalent bonds with more than one inhibitor molecule albeit in a much lower quantity.

## Specific binding of covalent compounds to CAIX expressed on live cell surface

The HeLa cell culture was grown under hypoxia and shown to express CAIX on the cell surface reaching the concentration of 2–10 nM, determined by saturating with fluorescein-labeled compound GZ19-32 as previously described (*Matulienė et al., 2022*). Covalent compounds were added to the cell culture at various concentrations together with 10 nM of GZ19-32 that strongly and specifically binds CAIX. The tested covalent compounds competed for the binding to the CAIX active site in a dose-dependent manner (*Figure 6*). At high concentrations (e.g. 100 μM, 10,000-fold surplus over CAIX and GZ19-32), the compounds completely outcompeted the CAIX-specific GZ19-32. However, at low concentrations, around 10 nM, the compounds competed with GZ19-32 depending on the compound's chemical nature. Thus, the covalent compounds were available for binding to CAIX and, most likely, did not bind to other proteins that are expected to be present in abundant quantities on the cell surface. These other proteins certainly have His residues that would have been modified if the non-specific binding occurred.

Several covalent compounds were chosen to demonstrate the importance of compound structural features for CAIX recognition in cell cultures. Two compounds **12** and **18** were *para*-substituted benzenesulfonamides, non-selective for CAIX. Compound **24** had a cyclododecyl amino substitution, selectively recognizing CAIX, but slightly too large for optimal binding and solubility. The **20** contains cyclooctylamine substitution at the *meta* position, exhibiting a high affinity for purified CAIX. Finally, **22** bore the cyclooctylamine substitution and the carbamate leaving group optimized for lower covalent modification activity compared to ester.

All tested covalent compounds competed with the fluorescein-labeled GZ19-32 for the binding to cell surface CAIX in a dose-dependent manner. Their apparent dissociation constants, as determined by the competition with GZ19-32, are listed in *Table 3*. The *para*-substituted compounds that are

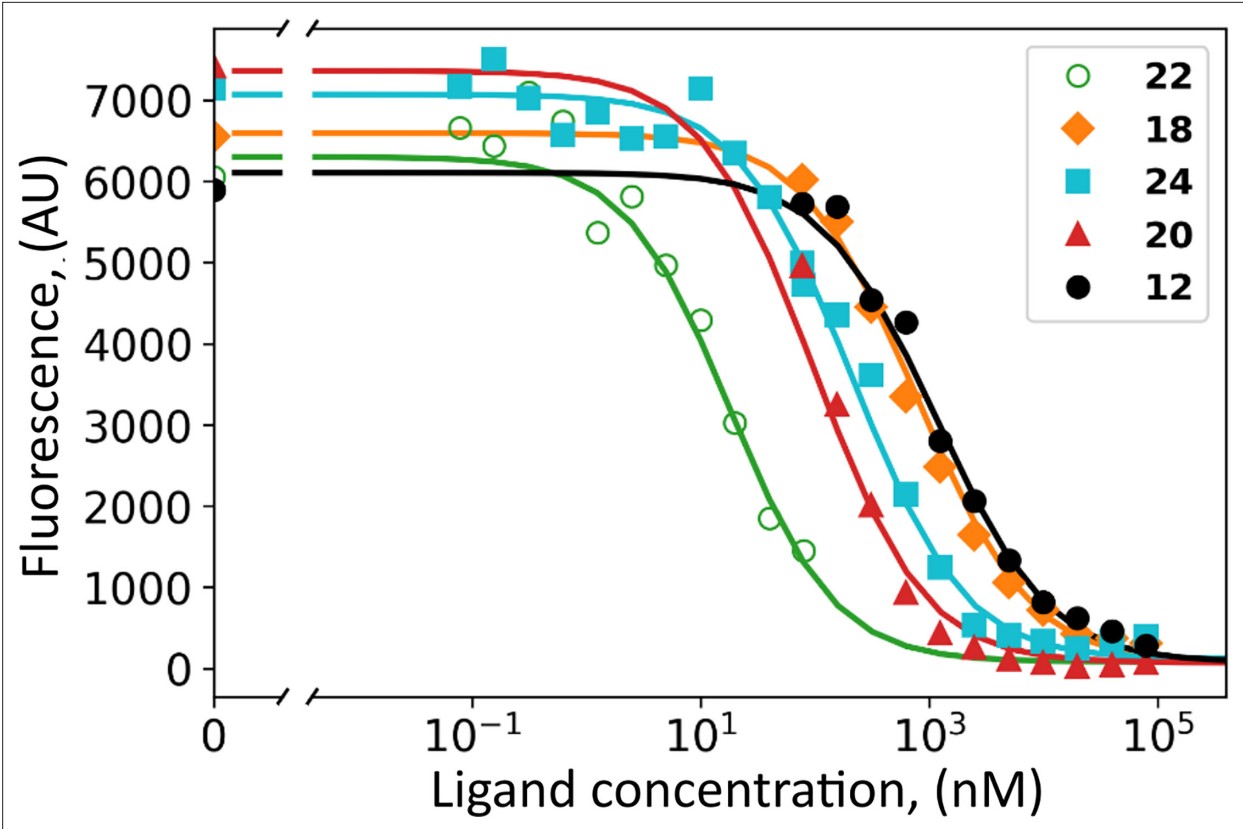

**Figure 6.** Dosing curves of covalent compounds applied to hypoxic live cell culture expressing CAIX: **22** – green; **18** - orange; **24** – cyan; **20** – red, and **12** – black. The compounds competed with the fluorescein-labeled GZ19-32, added at 10 nM concentration to all samples. A competitive binding model was applied to obtain the affinities of tested compounds for cell-surface CAIX (**Table 1**).

non-selective for CAIX, bound weaker to cell-surface CAIX than the *meta* substitute-bearing CAIX-selective compounds. The ester compounds bearing both *para* and *meta* substitutions designed for CAIX recognition exhibited single-digit nanomolar affinities (4.0 nM for **24** and 1.8 nM for **20**). However, carbamate compound **22** exhibited the strongest affinity (300 pM) for cell-surface CAIX among all tested compounds.

The carbamate compound **22** showed the highest affinity for cell-expressed CAIX and irreversibly covalently modified the protein in the active site, thus permanently inhibiting its enzymatic activity. Therefore, this compound is a leader among tested compounds to serve as an anticancer inhibitor of CAIX, highly expressed in hypoxic solid tumors.

**Table 3.** Affinities (apparent dissociation constants $K_{d,app}$) of covalent compound binding to cell-expressed CAIX determined by applying a competitive binding model to data in **Figure 6** as previously described (**Matulienė et al., 2022**).

Parameters used in the competitive model were: the CAIX protein concentration was 5 nM ($P_t$ = 5 nM), the dissociation constant of GZ19-32 was 150 pM ($K_{d\_B}$ = 150 pM), and the concentration of GZ19-32 was 10 nM ($L_{t\_B}$ = 10 nM).

| Compound | $K_{d,app\_A}$, nM |
|---|---|
| 12 | 22 |
| 18 | 15 |
| 24 | 4.0 |
| 20 | 1.8 |
| 22 | 0.30 |

## Covalent compound binding apparent affinities to purified CA isozymes

Covalent compounds formed an irreversible covalent bond with the protein molecule. This inhibition mode may occur in two stages. In the first stage, the inhibitor interacts with the enzyme due to its affinity to the targeted enzyme. Here, the affinity is determined by the primary sulfonamide group and the hydrophobic substituent in the *meta* position. The compound is still able to reversibly dissociate and its non-covalent binding affinity is quantified by the dissociation constant $K_d$, defined as the ratio of dissociation and association rate constants $k_{off}/k_{on}$:

$$\text{E} + \text{I} \underset{k_{off}}{\overset{k_{on}}{\rightleftharpoons}} \text{E} \cdot \text{I} \xrightarrow{k_{trans}} \text{E} \cdot \text{A} \xrightarrow{k_{inact}} \text{E} - \text{A}$$

In the second stage, the pre-vinylsulfone compound is chemically transformed into the reactive vinylsulfone electrophile by a basic amino acid of the enzyme at a rate of $k_{trans}$. In the final reaction step, the vinylsulfone may form a covalent bond with the nucleophilic residue with a specific inactivation rate constant $k_{inact}$. Since the vinylsulfone is highly reactive, $k_{trans}$ must be rate-limiting and accounting for the apparent inactivation rate, much slower than for vinyl sulfone **15**.

It is incorrect to state covalent compound affinities in terms of a conventional dissociation constant $K_d$. Therefore, the apparent dissociation constant is valid only to a limited extent because if there is an irreversible chemical modification, then eventually all of the protein will be modified independent of the affinity. In our case, we can assume a rapid pre-equilibrium followed by a slow covalent modification. Therefore, relative affinity measurements are valid both by competition assay described above and the fluorescence-based thermal shift assay, described below. However, due to the interplay of kinetic and thermodynamic equilibrium contributions, the affinity measurements should still be considered with caution.

We applied the fluorescence-based thermal shift assay to determine the apparent dissociation constants $K_{d,app}$ of covalent compounds to arrange them in the order of their apparent affinities (association rate constants). For example, the CAIX-specific covalent compound **22** bound with an extremely tight affinity, the apparent dissociation constant was determined to be 7.8 pM, the highest affinity among known CAIX-binding compounds. The thermal shift was over 18 °C and exhibited a typical flat dosing curve often observed for covalent compounds (*Figure 7*, *Figure 7—figure supplements 1–9*). There was a strong shift of the protein melting temperature caused by the compound but no further shift as observed in reversible non-covalent interactions. Using compound **18**, we determined that the obtained apparent affinity constants were time-independent, meaning lower errors due to compound-protein incubation time and preparation for FTSA (*Figure 7—figure supplements 10–15*).

The observed $K_{d,app}$ values of all covalent compounds binding to CA isozymes were higher than those of its non-covalent analogs. For example, CAIX binds to the *para*-substituted esters **12** and **14**, forming a covalent bond with the protein, up to 1000 times stronger than the non-covalent *para*-substituted compounds **1** and **3**. Covalent compounds with meta-substituents **20–24** bind CAIX up to 10 times more strongly than their non-covalent analogs **10** and **11**. The apparent dissociation constants of covalent compound binding to all 12 human catalytically active CA isozymes as determined by the fluorescence-based thermal shift assay are listed in *Table 4*. The apparent affinities of covalent compounds can be compared with the non-covalent reversibly-binding analog compounds and estimate the energetic contribution of the covalent bond.

## Design of dual CAIX and CAXII-recognizing covalent compounds

As shown above, compounds with a carbamate leaving group are better than ester groups. The carbamate compounds seemed to have a good balance to enhance interaction with CAIX via a covalent bond and, at the same time, should have sufficiently low reactivity to react with any unintended proteins. It has been demonstrated that, in some cancers, CAXII isozyme is overexpressed instead of CAIX and sometimes both of these isozymes are expressed. Therefore, a dual attack on CAIX and CAXII could be beneficial over a single CAIX interaction. At the same time, inhibition of the remaining 10 CA isozymes is expected to cause more harm than benefit.

As seen on the arrows going from two left non-covalent compounds to the adjacent covalent compounds (*Figure 8*), there is a significant gain in affinity due to the covalent bond, at least several hundred-fold stronger binding. Second, the presence of the cyclooctyl or cyclododecyl ring at the *meta*-position relative to sulfonamide increased the affinity for CAIX, but – what is even more

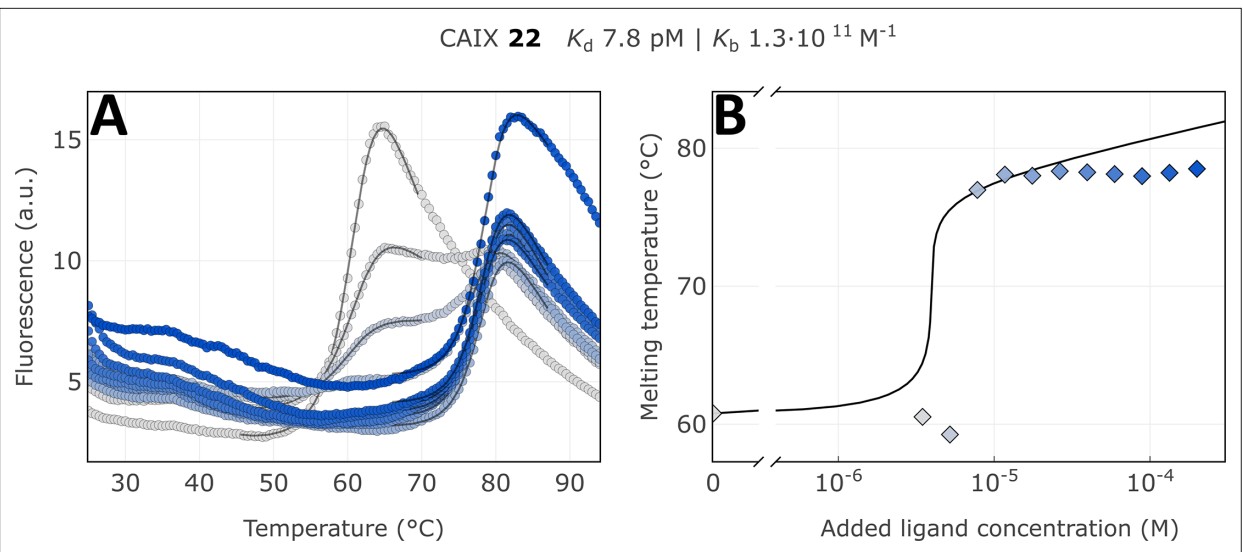

**Figure 7.** Apparent affinity determination of compound **22** by the thermal shift assay. (**A**) Raw FTSA data of compound **22** binding to CAIX (pH 7.0 for 37 °C). (**B**) Enzyme melting temperature dependence on compound **22** concentration. Datapoints saturated due to the covalent nature of interaction and therefore did not fully fit into the model line.

The online version of this article includes the following figure supplement(s) for figure 7:

**Figure supplement 1.** Raw FTSA data of compound **20** binding to CA I (**A**); enzyme melting temperature dependence on sulfonamide **20** concentration (**B**).

**Figure supplement 2.** Raw FTSA data of compound **20** binding to CA II (**A**); enzyme melting temperature dependence on sulfonamide **20** concentration (**B**).

**Figure supplement 3.** Raw FTSA data of compound **20** binding to CA IV (**A**); enzyme melting temperature dependence on sulfonamide **20** concentration (**B**).

**Figure supplement 4.** Raw FTSA data of compound **20** binding to CA VI (**A**); enzyme melting temperature dependence on sulfonamide **20** concentration (**B**).

**Figure supplement 5.** Raw FTSA data of compound **20** binding to CA VII (**A**); enzyme melting temperature dependence on sulfonamide **20** concentration (**B**).

**Figure supplement 6.** Raw FTSA data of compound **20** binding to CA IX (**A**); enzyme melting temperature dependence on sulfonamide **20** concentration (**B**).

**Figure supplement 7.** Raw FTSA data of compound **20** binding to CA XII (**A**); enzyme melting temperature dependence on sulfonamide **20** concentration (**B**).

**Figure supplement 8.** Raw FTSA data of compound **20** binding to CA XIII (**A**); enzyme melting temperature dependence on sulfonamide **20** concentration (**B**).

**Figure supplement 9.** Raw FTSA data of compound **20** binding to CA XIV (**A**); enzyme melting temperature dependence on sulfonamide **20** concentration (**B**).

**Figure supplement 10.** Raw FTSA data of compound **18** binding to CA I after a 2 hr incubation period (**A**); enzyme melting temperature dependence on sulfonamide **18** concentration (**B**).

**Figure supplement 11.** Raw FTSA data of compound **18** binding to CA I without incubating period (**A**); enzyme melting temperature dependence on sulfonamide **18** concentration (**B**).

**Figure supplement 12.** Raw FTSA data of compound **18** binding to CA II after a 2 hr incubation period (**A**); enzyme melting temperature dependence on sulfonamide **18** concentration (**B**).

**Figure supplement 13.** Raw FTSA data of compound **18** binding to CA II without an incubating period (**A**); enzyme melting temperature dependence on sulfonamide **18** concentration (**B**).

**Figure supplement 14.** Raw FTSA data of compound **18** binding to CA XIII after a 2 hr incubation period (**A**); enzyme melting temperature dependence on sulfonamide **18** concentration (**B**).

**Figure supplement 15.** Raw FTSA data of compound **18** binding to CA XIII without an incubating period (**A**); enzyme melting temperature dependence on sulfonamide **18** concentration (**B**).

**Table 4.** The apparent dissociation constants $K_{d,app}$ (in nM units) for compound interaction with human recombinant CA isozymes as determined by fluorescence-based thermal shift assay (FTSA) at pH 7.0 for 37 °C.
The values are logarithmic averages of several independent FTSA experiments.

| | Compound | $K_{d,app}$, nM | | | | | | | | | | | |
|---|---|---|---|---|---|---|---|---|---|---|---|---|---|
| | | CAI | CAII | CAIII | CAIV | CAVA | CAVB | CAVI | CAVII | CAIX | CAXII | CAXIII | CAXIV |
| | Non-covalent | | | | | | | | | | | | |
| 1 | VD10-35 | 0.2 | 20 | 20,000 | 500 | 300 | 20 | 70 | 7 | 40 | 300 | 30 | 30 |
| 2 | JA16-9-26 | 0.7 | 50 | 60,000 | 2000 | 600 | 5 | 400 | 20 | 5 | 500 | 100 | 20 |
| 3 | AV19-64-1 | 0.6 | 20 | 20,000 | 700 | 200 | 2 | 200 | 1 | 50 | 600 | 2 | 8 |
| 4 | AV21-05 | 1 | 10 | ≥200,000 | 700 | 70 | 8 | 300 | 3 | 20 | 200 | 10 | 10 |
| 5 | AV19-36 | 0.2 | 10 | 50,000 | 700 | 200 | 20 | 700 | 5 | 20 | 200 | 4 | 4 |
| 6 | AV20-08 | 60 | 70 | 2000 | 20 | 10 | 3000 | 1000 | 40 | 20 | 700 | 800 | 40 |
| 7 | AV19-56 | 0.3 | 20 | 20,000 | 400 | 300 | 20 | 200 | 20 | 20 | 100 | 20 | 9 |
| 8 | AV17-06 | 0.2 | 10 | 40,000 | 800 | 300 | 8 | 300 | 3 | 10 | 200 | 2 | 5 |
| 9 | AV23-30-1 | 0.03 | 2 | 7000 | 100 | nd | nd | 1000 | nd | 3 | 40 | 3 | 5 |
| 10 | AV17-30 | 4000 | 100 | ≥200,000 | 100 | 4000 | 20 | 400 | 30 | 0.1 | 10 | 20 | 40 |
| 11 | VD11-4-2 | 800 | 60 | 30,000 | 60 | 3000 | 20 | 70 | 9 | 0.03 | 3 | 4 | 4 |
| | Covalent | | | | | | | | | | | | |
| 12 | VD10-35-2 | 0.003 | 0.2 | 30,000 | 10 | 2 | 100 | 0.1 | 0.007 | 0.03 | 0.03 | 0.01 | 0.007 |
| 13 | VD18-02 | 50,000 | 300,000 | ≥1,000,000 | ≥1,000,000 | ≥1,000,000 | ≥1,000,000 | ≥1,000,000 | ≥1,000,000 | 300,000 | 300,000 | ≥1,000,000 | 30,000 |
| 14 | AV21-03 | 0.003 | 0.07 | 40,000 | 2 | 3 | 0.01 | 0.05 | 0.002 | 0.02 | 0.02 | 0.003 | 0.1 |
| 15 | AV22-48 | 0.007 | 0.3 | 30,000 | 10 | 1 | 0.01 | 0.3 | 0.006 | 0.07 | 0.03 | 0.06 | 0.06 |
| 16 | AV22-162 | 0.008 | 0.4 | 20,000 | 60 | nd | nd | 0.3 | 0.8 | 0.09 | 0.3 | 0.05 | 1 |
| 17 | AV21-25 | 0.002 | 0.1 | 20,000 | 50 | 3 | 0.02 | 0.4 | 0.8 | 0.04 | 0.07 | 0.01 | 0.3 |
| 18 | AV22-138 | 0.008 | 0.4 | 4000 | 20 | 1 | 0.005 | 0.1 | 0.4 | 0.04 | 0.1 | 0.02 | 0.4 |
| 19 | AV23-04 | 0.003 | 0.3 | nd | 20 | nd | nd | 0.06 | 0.4 | 0.04 | 0.06 | 0.02 | 0.5 |
| 20 | AV19-37 | 0.05* | 0.8 | ≥200,000 | 1 | 500 | 4 | 10 | 0.04 | 0.004 | 0.02 | 0.02 | 0.2 |
| 21 | AV21-08 | 0.005* | 0.7 | ≥200,000 | 2 | 1000 | 0.02 | 3 | 0.2 | 0.002 | 0.01 | 0.005 | 0.3 |
| 22 | AV22-149 | 0.01* | 1 | 100,000 | 2 | nd | 0.1 | 40 | 80 | 0.008 | 0.01 | 0.02 | 2 |
| 23 | AV21-47 | 10* or 0.02 | 1 | ≥200,000 | 0.2 | 5000 | 0.3 | 30 | nd | 0.003 | 0.1 | 0.02 | 0.5 |
| 24 | AV22-132 | 20* or 0.02 | 2 | ≥200,000 | 4 | 3000 | 0.2 | 2000 | nd | 0.009 | 0.2 | 0.01 | 0.9 |

*2 different fluorescence shifts were observed of which the dominating one was used for $k_{d,app}$ determination.

significant – greatly reduced compound affinity for non-target CAI and CAII. The covalent CAIX-targeting compounds reached the affinity of single-digit picomolar, an incredibly high value, never reached by any CAIX-binding compounds and rare among any interactions. Despite the apparent selectivity of compound **23** to CAIX compared to CAI, compound **22** is more promising as a drug candidate due to its slower covalent bond formation rate and lower associated off-target toxic effects.

## Discussion

In this work, we are introducing a novel pre-vinylsulfone warhead for targeted covalent modification of proteins. The designed group of compounds bound to CAIX, an anti-cancer target protein, via a triple binding model: (1) the sulfonamide group formed a coordination bond with the Zn(II) in the active site, (2) the hydrophobic ring selectively recognized CAIX over other CA isozymes, and (3) covalent bond formed between the compound and histidine residue of the protein. To reduce reactivity, compounds with the carbamate leaving group were designed. A large series of synthesized

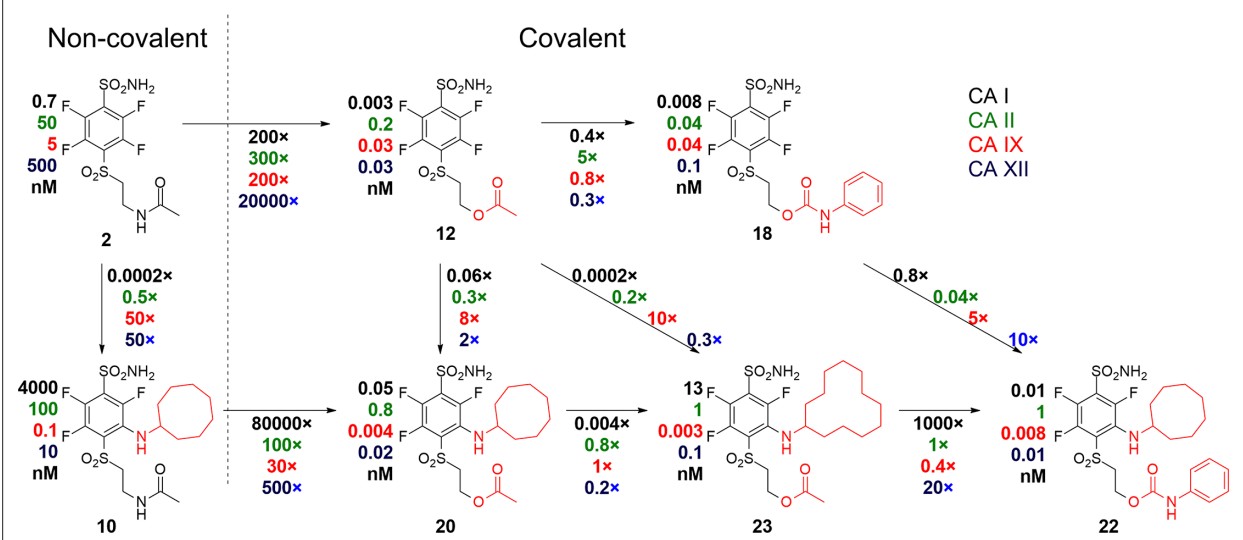

**Figure 8.** A correlation map between chemical structures and binding affinities showing apparent dissociation constants of compounds for CAI, CAII, CAIX, and CAXII, in nM units. Apparent affinities are listed next to compound structures and the ratios of $K_{d,app}$ – above or below the arrows connecting compounds that are compared. The two left compounds, both upper and lower, located to the left of the vertical dashed line do not form covalent bonds with the proteins, while the rest of the compounds form the covalent bond.

compounds distinguish the chemical structures necessary for covalent modification from non-covalent reversible interaction with the protein.

In recent years, covalent inhibitors gained much attention due to their advantages such as complete target protein inhibition, low dosage, and effectiveness against mutated targets (unless mutation happens in the targeted nucleophilic residue; *Lonsdale and Ward, 2018*; *Baillie, 2016*; *Kim et al., 2021*). Although there is still a lot of concern for covalent compounds' off-target toxic effects, success stories like ibrutinib or, almost a century ago discovered penicillin and aspirin prove that carefully designed covalent compounds can be safely used (*Byrd et al., 2014*; *Singh et al., 2011*). To this day, the most used strategy in the design of covalent inhibitors is by attaching an optimized 'warhead' to a known lead compound. Such inhibitors exhibit efficient and full target inactivation. Quite a few known electrophilic groups are acting as efficient covalently modifying warheads. However, it is necessary to choose electrophiles with balanced reactivity to avoid off-target toxicity while maintaining steady covalent bond formation with targeted amino acid residue. Thus, among many discovered warheads only six are FDA-approved (*De Vita, 2021*). To the best of our knowledge, the fragment $SO_2CH_2CH_2OCOR$ has never been previously described in the literature as a precursor of warhead, a pro-drug capable of rearranging to vinyl-sulfonyl moiety and forming a covalent bond with the target enzyme.

Nevertheless, it is known that compounds bearing $SO_2CH_2CH_2OCOR$ fragments can rearrange to vinyl-sulfonyl moiety which has been known as a covalently modifying warhead by reacting with lysine/cysteine residues (*De Vita, 2021*). In most cases, it was demonstrated by chemical reactions in basic environments (*Tsutsui et al., 1987*; *Alonso et al., 2003*; *Ichikawa et al., 1989*). However, here we demonstrate that the vinyl-sulfonyl group can react with the His residue. Formation of the covalent bond was shown by X-ray crystallography and 2D NMR. The inability to regain enzymatic activity by dialysis of the compound also confirmed the covalent bond formation.

Several methods exist for comparing the affinity of covalent inhibitors for a target protein, such as comparing compound $K_d$, $IC_{50}$, or $K_i$ (*Lonsdale and Ward, 2018*; *Ruddraraju and Zhang, 2017*). However, due to the time-dependent nature of covalent inhibitor action, the conventional comparisons become challenging or even impractical using these parameters (*Lonsdale and Ward, 2018*; *Ruddraraju and Zhang, 2017*). In the case of $IC_{50}$, this value represents the compound concentration inhibiting half of the target enzyme molecules. However, in the case of covalent inhibitors that react irreversibly but slowly, given enough time, covalent inhibitors should give $IC_{50}$ values equal to half of the target concentration as a result of disrupted binding equilibrium (*Strelow, 2017*). The same

principle applies to $K_i$, where if covalent bond formation outpaces compound dissociation, leading to near zero $k_{off}$, the observed $K_i$ values should also approach zero over time. Thus, the $K_i$ alone is insufficient, since it does not take into account the second stage involving covalent bond formation. In a covalent inhibition model, where initial non-covalent binding precedes covalent bond formation, the most accepted way of describing covalent inhibitor binding commonly involves using $k_{inact}/K_i$ or % covalent occupancy derived from covalent kinetics and pharmacokinetics (*Strelow, 2017*). This approach, however, has limitations, especially with compounds exhibiting extremely high picomolar apparent affinities. We propose and have demonstrated that the thermal shift assay could be employed for the precise determination of such affinities.

Considering the already complicated covalent inhibitor evaluation due to two-step mechanism, it is even harder to assess compounds bearing $SO_2CH_2CH_2OCOR$ fragments because they act as prodrugs. We must consider an additional step – the elimination reaction, during which an active compound is formed, capable of binding to the protein covalently. Without the elimination step, the active compound is not formed, and the formation of a covalent bond with CA is impossible. If the elimination reaction rate is higher than the covalent bond formation rate, we can ignore it and consider it as part of $k_{inact}$ because the limiting step will be $k_{inact}$. However, it is challenging to determine separate and correct $K_i$, elimination rate constant, and $k_{inact}$ values for CA isozymes because of exceptionally high compound affinity. Nevertheless, using fluorescent thermal shift assay (FTSA), we could determine the $K_{d,app}$ for all 12 catalytically active CA isozymes and could perform an affinity correlation between different CA isozymes. Prior applications of the thermal shift were limited to testing the change in the melting temperature of the protein upon covalent modification (*Udompholkul et al., 2023*; *Zhao et al., 2021*; *Iliev et al., 2022*; *Qiao et al., 2023*; *Gambini et al., 2019*). Therefore, to the best of our knowledge, the apparent affinity determination by FTSA of covalent compound binding to proteins is being demonstrated here for the first time.

There are only a few examples of covalent inhibitors of CA – bromoacetazolamide and N-bromoacetylacetazolamide and several compounds designed for enzyme tagging which have been tested on CAII as a model enzyme (*Kandel et al., 1970*; *Chen et al., 2003*; *Takaoka et al., 2006*). In bromoacetazolamide case, even with 20 molar surpluses, bovine CAII was not fully modified even after 24 hr (*Takaoka et al., 2006*). The experiments with bromoacetazolamide were performed in the basic environment (pH 8.2 and 8.7), which was a favorable condition for covalent bond formation and thus it is hard to compare with other compounds (*Kandel et al., 1968*). The covalent tag-bearing vinyl-sulfonyl warhead showed better results compared to bromoacetazolamide (>90% of compound covalently bound to bovine CAII after 10 hr at pH 7.4). The compounds **20** and **12** irreversibly bound and inhibited most of the human CA isoforms in less than 2 hr (at pH 7.0) and possibly are the first covalent CA inhibitors tested against CA isozymes, except the above-described inhibitors of CAII and CAI, with vinyl-sulfonyl reacting with the histidine residue (*Chen et al., 2003*).

To assess the potential of covalent compounds for drug development, it is crucial to investigate whether they can indiscriminately react with different histidine and nucleophilic groups in proteins. Our testing involved examining the binding of these compounds to live cancer cells, thereby interacting with all proteins exposed on the cancer cell surface. The results demonstrated that these compounds exhibited specific and exclusive binding to the target protein CAIX, expressed in hypoxia-grown cancer cells. Notably, these compounds displayed the highest affinities among numerous CAIX inhibitors described in the literature, and their covalent binding led to irreversible inactivation of CAIX expressed in live cancer cells. This compelling evidence suggests significant potential for the development of these compounds as anticancer drugs.

# Materials and methods
## Enzyme purification

All recombinant human CA isozymes were produced as described previously by using either bacterial or mammalian expression system (*Dudutienė et al., 2014*; *Mickevičiūtė et al., 2019*). For isozymes possessing transmembrane parts, only catalytic domains were produced. The production of the catalytic domain of CAIX in methylotrophic yeast *Pichia pastoris* was performed as described in *Leitans et al., 2015*. Protein purity was confirmed by SDS-PAGE, and MW was confirmed by mass spectrometry.

## Fluorescent thermal shift assay (FTSA)

Thermal unfolding experiments of the purified CA isozymes were carried out by a real-time PCR instrument, Rotor-Gene Q, containing six channels. Twofold serial dilutions of the 10 mM compound stock in DMSO were made by adding 10 µL of DMSO to 10 µL of each compound solution. Overall, 8 different compound concentration solutions were prepared, including the 10 mM compound stock concentration and a sample containing only DMSO, without a ligand. To prepare 12 different concentrations of the ligand, 1.5-fold serial dilutions of 10 mM compound stock were performed by adding 10 µL of DMSO to 20 µL of each compound solution (the last sample contained no ligand). Each prepared compound solution was diluted 12.5 times with the assay buffer 50 mM sodium phosphate (pH 7.0), 100 mM NaCl. The 10 µM CA isozyme solution (or 20 µM of CAIV) was prepared in the same assay buffer, which contained a reporter dye (200 µM ANS or 200 x diluted Glomelt). 5 µL of prepared CA solution with dye was added to the 100 µl of PCR tubes. Subsequently, 5 µL of the compound solution is added. The tubes are placed into the real-time thermocycler, and protein unfolding is measured by increasing the temperature from 25°C to 90°C, at the rate of 1 °C/min, and measuring the fluorescence of the dye. The raw data were analyzed to determine the $T_m$ of the proteins. $T_m$ values were plotted as a function of ligand concentration and the model was fitted to the dosing curves to obtain the binding affinities using Thermott software (*Gedgaudas et al., 2022*). Compound binding affinity and other thermodynamic data will be entered and freely available for download from the repository plbd.org (Protein-Ligand Binding Database; *Lingė et al., 2023*).

## CAIX activity measurement by the stopped-flow assay

CAIX activity was measured in the absence and presence of the compound before and after dialysis. 1.5 µM CAIX was incubated with 15 µM **20**, 15 µM **22**, or 100 µM **5** for 2 hr. The solution of CAIX without a compound and the CAIX-compound complexes were then dialyzed in 25 mM Tris buffer solution (pH = 7) containing 50 mM NaCl (by changing buffer four times every 8/16 hours). $CO_2$ hydration velocities were measured by recording the absorbance of phenol red (final concentration 50 µM) at 557 nm using an Applied Photophysics SX.18MV-R stopped-flow spectrometer. Experiments were performed at 25 °C using 25 mM Hepes containing 50 mM NaCl, pH 7.5.

## Mass spectrometry

Mass spectrometry experiments were performed with an electrospray ionization time-of-flight mass spectrometer (Q-TOF). The 0.1 mg/mL CA isozyme solution was prepared in the absence or presence of compounds (1:2; 1:5, or 1:10 CA isozyme: compound molar ratio). The solution was incubated for 1 hr at room temperature before analysis. The final DMSO concentration was 1% (v/v).

## 2D NMR

All NMR spectra were recorded using a 600 MHz Bruker Avance Neo spectrometer equipped with a cryogenic probe. 2D $^{15}N$–$^1H$ HSQC spectra of $^{15}N$ labeled CAII solution (270 µM 15 N CAII, 20 mM sodium phosphate buffer, 50 mM NaCl, 5% $D_2O$, pH 6.8) were recorded at 25 °C using 256 increments in the indirect dimension and 8 scans. The spectra were recorded when the protein solution contained different concentrations of covalent ligand: 0.27 mM, 0.53 mM, 1.0 mM and 1.5 mM (the final DMSO concentration was 7.5%) or non-covalent ligand: 0.35 mM, 0.70 mM, 1.0 mM, 2.0 mM (the final DMSO concentration was 3%). The spectra were analyzed using Topspin 4.1.3 and CcpNMR 2.4.1 analysis software (*Vranken et al., 2005*).

## HPLC analysis

The HPLC separation was done as previously described (*Sakalauskas et al., 2022*). In brief, the samples were separated and analyzed using Shimadzu UFLC system with a CMB-20A communication module, two LC20AD quaternary and isocratic pumps, a SIL-20AC autosampler, a CTO-20A column compartment and an SPD-M20A DAD detector (Shimadzu Corp., Japan). For the detection of the eluting molecules, the DAD spectra recording was set from 190 to 750 nm with a data rate of 6.25 Hz. The ACE C18-PFP HPLC separation column (100x4.6, 3 µm, Avantor) was used as a stationary phase. The HPLC grade MeCN (Fisher Scientific) and Milli-Q water (18.2 MΩ cm$^{-1}$, Milli-Q Plus system, Millipore Bedford, MA, USA) were used for the RP-HPLC separation.

The samples were separated using a trinary gradient consisting of ultrapure water (eluent A), MeCN (eluent B), and 1% TFA in ultrapure water (eluent C). A constant 10% flow of eluent C was used to maintain 0.1% TFA concentration in the column throughout the separation experiment. The gradient between eluents A and B was 36% (0 min), 63% (20 min), and 63% (21 min). Before each analytical run, the column equilibration (10 column volume) was performed. The column thermostat was set to 40 °C and the flow rate to 1 mL min$^{-1}$.

## Protein-compound complex sample preparation for data-dependent analysis (DDA)

The 0.1 mg/ml CAXIII solution was prepared with compound **12** (molar ratio 1:2 CAXIII: **12**) and filter-aided sample preparation (FASP) (*Wiśniewski et al., 2009*) was used for protein digestion before mass spectrometry analyses.

## Data-dependent analysis

DDA was performed with the nanoAcquity coupled to a Synapt G2 HDMS mass spectrometer (Waters). For DDA, the instrument performed a 0.7 s MS scan (350–1350 scan range) followed by MS/MS acquisition on the top 5 ions with charge states 2+, 3+, and 4+. MS/MS scan range was 50–2000 Da, 0.6 s scan duration with exclusion after 2 MS/MS scans were acquired, and dynamic exclusion of ions within 100 mDa of the selected precursor m/z was set to 100 s.

The Progenesis QI for proteomics software (Nonlinear Dynamics), in combination with the Mascot server (2.2.07) was employed to identify peptides. The acquired raw files were imported into the Progenesis QI for proteomics software and MS2 spectra were exported directly from Progenesis in mgf format and searched using the MASCOT algorithm.

## Crystallization and structure solution

Human CAI in buffer containing 20 mM HEPES pH 7.6 and 50 mM NaCl was mixed with the **21** in DMSO in the ratio 1:1.1 and incubated for 2.5 hr at room temperature. The protein-**21** complex was concentrated at 30 mg/ml, and crystallization in sitting drops was started. The crystallization solution contained 0.1 M Tris-HCl pH 8.5, 0.2 M NaCl, and 28% (w/v) PEG3350. Before cryo-cooling crystal was shortly incubated in cryo-protection buffer containing 0.1 M TrisHCl pH8.5, 15% (w/v) PEG8000, and 20% (v/v) Ethylene glycol. The synchrotron data was collected at beamline P13 operated by EMBL Hamburg at the PETRA III storage ring (DESY, Hamburg, Germany).

Diffraction data were integrated by XDS (*Kabsch, 2010*), scaled using AIMLESS 0.7.4 and other CCP4 tools v. 7.1.002 (*Agirre et al., 2023*). The structure was solved by molecular replacement using MOLREP v.11.7.02 (*Vagin and Teplyakov, 2010*) and 2CAB as an initial model. The model was refined by REFMAC v. 5.8.0258 (*Murshudov et al., 2011*) and rebuilt in COOT v.0.9 (*Emsley et al., 2010*). Inhibitor model was created and minimized using AVOGADRO v. 1.2.0 (*Hanwell et al., 2012*).

CAII protein was concentrated at 10 mg/ml in a 20 mM Tris-HCl buffer. It was then mixed with compound **20** (5 mM final concentration) and incubated overnight at 4 °C. Previously known crystallization conditions for CAII did not yield any crystals when co-crystallizing with compound **20**. Crystallization condition screening was performed using the Morpheus screen from Molecular Dimensions. After slight optimization, crystals were grown using the sitting drop technique in 0.06 M magnesium chloride hexahydrate; 0.06 M calcium chloride dihydrate; 0.1 M Tris pH 8.5; 20% v/v PEG 500 MME; 10% w/v PEG 20000 conditions diffracted at 1.4 Å resolution. The alteration in crystallization conditions was likely due to the formation of a covalent bond between the ligand and the enzyme. The dataset of the CAII-ligand complex was collected at BESSY II beamline 14.1 and processed using MOSFLM (*Battye et al., 2011*) and SCALA (*Evans, 2006*). Molecular replacement was performed using MOLREP (*Vagin and Teplyakov, 2010*) with 5AMD (*Ivanova et al., 2015*) as the initial model.

CAIX protein was concentrated to 10 mg/ml in a 20 mM Tris-HCl buffer. It was then mixed with compound **23** (5 mM final concentration) and incubated overnight at 4 °C. Crystals grew using similar co-crystallization conditions as described before (*Leitans et al., 2015*).

The dataset of the CAIX-ligand complex was collected at the Diamond Light Source beamline I03 and processed using XDS (*Kabsch, 2010*) and AIMLESS (*Agirre et al., 2023*). Molecular replacement was performed using MOLREP (*Vagin and Teplyakov, 1997*) with 8Q18 (*Leitans et al., 2023*) as the initial model.

Model refinement was performed with REFMAC (*Murshudov et al., 2011*), and the structures were visualized using COOT (*Emsley et al., 2010*). Ligand parameter files were generated using LIBCHECK (*Lebedev et al., 2012*), and the ligand was manually fitted to the electron density map in COOT (*Emsley et al., 2010*). The coordinates and structure factors have been deposited in the PDB. Table ST1 provides the PDB access code, data collection, and refinement statistics.

## Determination of compound $K_d$ values for cellular CAIX

Human cervical adenocarcinoma cells (HeLa) were cultured in Dulbecco's Modified Eagle's Medium (DMEM) with GlutaMAX (Gibco, ThermoFisher) supplemented with 10% fetal bovine serum (ThermoFisher) in a humidified atmosphere at 37 °C and 5% $CO_2$.

A covalent compound competition experiment with fluorescein-labeled compound GZ19-32 was conducted as described previously (*Matulienė et al., 2022*). In brief, HeLa cells were cultivated in DMEM in 12-well plates under hypoxic conditions (1% $O_2$) for 72 hr. The 10 serial twofold dilutions of covalent compounds were prepared in FluoroBrite DMEM (ThermoFisher) starting with 80 nM (1st tube). No compound was added to the last 12th tube (it contained FluoroBrite only). Subsequently, the same volume of 20 nM GZ19-32 was added to each of the 12 tubes and mixed. Cell culture medium was removed from all 12 wells with HeLa cells and 200 μl of prepared compound mixtures were added, followed by 20 min incubation at 37 °C under normoxic conditions (21% $O_2$). Post-incubation, the compound solutions were aspirated, and the cells were washed 3 times with 400 μl of PBS. Finally, the cells were detached from the well plate surface by TrypLE (ThermoFisher), resuspended by pipetting in 200 μl of FluoroBrite DMEM, and 150 μl of the suspension was transferred to black Thermo Scientific Nunc MicroWell 96-Well Optical-Bottom Plates for fluorescence and absorbance measurements.

## Compound synthesis

Compound numbering includes the intermediate compounds, and, therefore, there are two numbering systems. Compounds are marked with x to help visualize the path of synthesis and the numbered compounds are listed in the brackets. Chemical compounds may be provided upon reasonable request for research purposes if available in stock. Compound synthesis paths are shown in *Figures 9–11*.

All starting materials and reagents were commercially available or prepared according to known procedures. Melting points of the compounds were determined in open capillaries on a Thermo Scientific 9100 Series and are uncorrected. $^1$H and $^{13}$C NMR spectra were recorded on a Bruker spectrometer (400 and 100 MHz, respectively) in DMSO-$D_6$ or CDCl$_3$ using residual DMSO, CDCl$_3$ signals (2.50 ppm, 7.26 ppm, and 39.52 ppm, 77.16 ppm for $^1$H and $^{13}$C NMR spectra, respectively) as the internal standard. $^{19}$F NMR spectra were recorded on a Bruker spectrometer (376 MHz) with CFCl$_3$ as an internal standard. TLC was performed with silica gel 60 F254 aluminum plates (Merck) and visualized with UV light. Column chromatography was performed using silica gel 60 (0.040–0.063 mm, Merck). High-resolution mass spectra (HRMS) were recorded on a Dual-ESI Q-TOF 6520 mass spectrometer (Agilent Technologies). Compound IUPAC names were generated with ChemDraw ultra 12.0.

2,3,4,5,6-pentafluorobenzenesulfonamide (**2ax**) was prepared according to known procedure in the literature (*Dudutienė et al., 2013*). To a 100 ml cooled to –10 °C temperature THF, 2,3,4,5,6-pentafluorobenzenesulfonyl chloride (**1x**) (5 ml; 33.70 mmol; 1 eq.) was dissolved and NH$_3$ (10 %) (10 ml) was slowly added dropwise while mixing as well as keeping reaction temperature under –10 °C. After reaction completion mixture was basified to pH 8–9 and left stirring for an additional hour at room temperature. The product was recrystallized from H$_2$O and white crystals were obtained. Yield: 5.33 g; (64%). Mp: 154–155°C (close to the value in the literature [*Dudutienė et al., 2013*], mp: 155–156°C).

2,3,4,5,6-pentafluoro-N-methylbenzenesulfonamide (**2bx**) synthesis was described in *Baronas et al., 2021*. To a 100 ml cooled to –10 °C temperature THF, 2,3,4,5,6-pentafluorobenzenesulfonyl chloride (**1x**) (2 ml; 13.47 mmol; 1 eq.) was dissolved and 2 M methylamine in methanol (0.837 g; 26.9 mmol; 2 eq.) was slowly added dropwise while mixing as well as keeping reaction temperature under –10 °C. The reaction process was monitored using TLC and after reaction completion, THF was evaporated under reduced pressure. The product was recrystallized from MeOH:H$_2$O (1:4) and white crystals were obtained. Yield: 2.6 g; (74%). Mp: 95–97°C (close to the value in the literature [*Baronas et al., 2021*], mp: 96–97°C).

2,3,5,6-Tetrafluoro-4-((2-hydroxyethyl)thio)benzensulfonamide (**3ax**) was prepared according to the known procedure in literature (*Dudutienė et al., 2013*). 2,3,4,5,6-pentafluorobenzensulfonamide

Reagents: **a)** NH$_3$ (25%), THF (**2ax**); NH$_2$CH$_3$ (2M in MeOH), THF (**2bx**); **b)** 2-mercaptoethan-1-ol, Et$_3$N, MeOH (**3ax, bx**); 3-mercaptopropan-1-ol, Et$_3$N, MeOH (**3cx**); methyl 3-mercaptopropanoate, Et$_3$N, MeOH (**6x**); 2-mercapto-2-methylpropan-1-ol, Et$_3$N, MeOH (**12x**), N-(2-mercaptoethyl)acetamide, Et$_3$N, MeOH (**16x**); **c)** H$_2$O$_2$ (30%), acetic acid, 75 °C; **d)** acetic acid, toluene, H$_2$SO$_4$ (conc.) (**5ax-cx**); propionic acid, toluene, H$_2$SO$_4$ (conc.) (**5dx**); pivalic acid, toluene, H$_2$SO$_4$ (conc.) (**5ex**); phenylacetic acid, toluene, H$_2$SO$_4$ (conc.) (**5fx**); **e)** H$_2$O$_2$ (30%), acetic acid, room temperature; **f)** thionyl chloride, Et$_3$N, MeCN.

\* Compounds **5bx** and **5cx** were obtained and purified during compound **3bx** and **3cx** oxidation reaction.

**Figure 9.** Synthesis paths of compounds 5ax-fx, 7x, 8x, 10x, 11x, 14x, and 16x.

**Figure 10.** Synthesis path of compound **21x**.

(**2ax**) (1.000 g; 4.05 mmol; 1 eq.), 2-mercaptoethanol (0.341 ml; 4.86 mmol; 1.2 eq.), Et$_3$N (0.677 ml; 4.86 mmol; 1.2 eq.) were dissolved in MeOH (20 ml) and left to stir at room temperature overnight. Next morning additional 2-mercaptoethanol (0.085 ml; 1.21 mmol; 0.3 eq.) and Et$_3$N (0.169 ml; 1,21 mmol; 0.3 eq.) portions were added and the reaction mixture was left to stir for 2 hours. After reaction completion, the solvent was evaporated under reduced pressure and the resultant precipitate was washed with H$_2$O. Recrystallization was accomplished from H$_2$O. Yield: 1.029 g; (83%). Mp: 111–112°C (close to the value in the literature [**Dudutienė et al., 2013**], mp: 111–112°C).

2,3,5,6-Tetrafluoro-4-((2-hydroxyethyl)thio)-N-methylbenzenesulfonamide (**3bx**) was prepared according to the known procedure in the literature (**Baronas et al., 2021**). 2,3,4,5,6-Pentafluoro-N-m ethylbenzenesulfonamide (**2bx**) (0.500 g; 1.92 mmol; 1eq.), 2-mercaptoethanol (0.175 ml; 2.50 mmol; 1.3 eq.), Et$_3$N (0.349 ml, 2.50 mmol, 1.3 eq.) were dissolved in MeOH (20 ml) and left to stir at room temperature for 20 hr. After reaction completion, the solvent was evaporated under reduced pressure and the resultant precipitate was washed with H$_2$O. Recrystallization was accomplished from H$_2$O. Yield: 0.464 g; (76%). Mp: 99 °C. $^1$H NMR (400 MHz, DMSO-d$_6$, δ): 2.62 (3H, s, SO$_2$NHC$\underline{H}_3$) 3.15 (2H, t, J=6.1 Hz, SCH$_2$), 3.59 (2H, q, J=5.9 Hz, CH$_2$O), 4.94 (1H, t, J=5.4 Hz, OH), 8.42 (1H, s, NH). $^{13}$C NMR (100 MHz, DMSO-d$_6$, δ): 28.28 (SO$_2$NHCH$_3$), 36.52 (SCH$_2$, t, J ($^{19}$F – $^{13}$C)=3.0 Hz), 60.66 (CH$_2$O), 118.23 (C1, t, J ($^{19}$F – $^{13}$C)=15.8 Hz), 119.99 (C4, t, J ($^{19}$F – $^{13}$C)=20.2 Hz), 143.01 (C2 and C6, ddt, $^1$J ($^{19}$F – $^{13}$C)=253.7 Hz, $^2$J ($^{19}$F – $^{13}$C)=16.9 Hz, $^3$J ($^{19}$F – $^{13}$C)=4.5 Hz), 146.41 (C3 and C5, ddt, $^1$J ($^{19}$F – $^{13}$C)=243.8 Hz, $^2$J ($^{19}$F – $^{13}$C)=15.4 Hz, $^3$J ($^{19}$F – $^{13}$C)=3.9 Hz). $^{19}$F NMR (376 MHz, DMSO-d$_6$, δ): –132.79 to -132.97 (2F, m), –139.08 to -139.25 (2 F, m).

2,3,5,6-tetrafluoro-4-((3-hydroxypropyl)thio)benzenesulfonamide **3cx**. 2,3,4,5,6-pentafluorbenzens ulfonamide (**2ax**) (0.400 g; 1.62 mmol), 3-mercaptopropan-1-ol (0.167 ml; 1.94 mmol; 1.2 eq.), Et$_3$N (0.270 ml; 1.94 mmol; 1,2 eq.) were dissolved in MeOH (15 ml) and left to stir at room temperature over night. Next morning additional 3-mercaptopropan-1-ol (0.027 ml; 0.32 mmol; 0.2 eq.) and Et$_3$N (0.044 ml; 0.32 mmol; 0.2 eq.) portions were added and reation mixture was left to stir for 3 hr. After reaction completion the solvent was evaporated under reduced pressure and product was purified by column chromatography silica gel, EtOAc/CHCl$_3$ (1:1), Rf = 0.39. Yield 0.386 (75%). Mp: 148–149°C. $^1$H NMR (400 MHz, DMSO-d$_6$, δ): 1.65 (2H, p, J=6.3 Hz, CH$_2$C$\underline{H}_2$CH$_2$), 3.09 (2H, t, J=7.3 Hz, CH$_2$S), 3.46 (2H, t, J=6.0 Hz, CH$_2$O), 8.39 (2H, s, SO$_2$NH$_2$). $^{13}$C NMR (100 MHz, DMSO-d$_6$, δ): 30.77 (SCH$_2$, t, J ($^{19}$F-$^{13}$C)=3.1 Hz), 32.68 (CH$_2$C$\underline{H}_2$CH$_2$), 58.70 (CH$_2$O), 118.67 (C1, t, J ($^{19}$F – $^{13}$C)=20.4 Hz), 122.46 (C4, t, J ($^{19}$F – $^{13}$C)=15.7 Hz), 142.54 (C2 and C6, ddt, $^1$J ($^{19}$F – $^{13}$C)=253.6 Hz, $^2$J ($^{19}$F – $^{13}$C)=16.9 Hz, $^3$J ($^{19}$F – $^{13}$C)=4.3 Hz), 146.48 (C3 and C5, ddt, $^1$J ($^{19}$F – $^{13}$C)=245.3 Hz, $^2$J ($^{19}$F – $^{13}$C)=13.9 Hz, $^3$J ($^{19}$F – $^{13}$C)=3.8 Hz). $^{19}$F NMR (376 MHz, DMSO-d$_6$, δ): –133.23 to -133.38 (2F, m), –139.04 to -139.19 (2 F, m). HRMS for C$_9$H$_9$F$_4$NO$_3$S$_2$ [(M+H)$^+$]: calc. 320.0033, found 320.0028.

**Figure 11.** Synthesis paths of compounds 24 ax, 24bx, 27 ax, 27bx, 28 ax, 28bx, 30 x, and 31 x.

2,3,5,6-Tetrafluoro-4-((2-hydroxyethyl)sulfonyl)benzenesulfonamide (**4ax**) was prepared according to the known procedure in the literature (**Dudutienė et al., 2013**). 2,3,5,6-Tetrafluoro-4-((2-hydroxyet hyl)thio)benzensulfonamide (**3ax**) (1.542 g; 5.06 mmol; 1 eq.) was dissolved in acetic acid (30 ml) and heated at 75 °C temperature for 18 hr. $H_2O_2$ (30%) was added in portions (0.1 ml) every 30 min (overall 3.6 ml) until complete starting material conversion. Afterward, the solvent was evaporated under reduced pressure and the product was purified by column chromatography (silica gel, EtOAc/CHCl$_3$ (1:1), Rf = 0.30). Yield: 0.768 g; (45%). Mp: 138–139°C (close to the value in the literature [**Dudutienė et al., 2013**], mp: 139–140°C).

Compounds **5bx** and **4bx** were synthesized during compound **3bx** oxidation.

2,3,5,6-Tetrafluoro-4-((2-hydroxyethyl)thio)-N-methylbenzenesulfonamide (**3bx**) (0.300 g, 0.94 mmol) was dissolved in acetic acid (10 ml) and heated at 75 °C temperature for 21 hr. $H_2O_2$ (30%) was added in portions (0.1 ml) every hour (overall 2.1 ml) until complete starting material conversion. Afterward, the solvent was evaporated under reduced pressure and products were purified by column chromatography (silica gel, EtOAc/CHCl$_3$ (1:1), Rf$_{4bx}$=0.53, Rf$_{5bx}$=0.79).

2,3,5,6-Tetrafluoro-4-((2-hydroxyethyl)sulfonyl)-N-methylbenzenesulfonamide **(4bx)**. Yield: 0.119 g; (36%). Mp: 201 °C. $^1$H NMR (400 MHz, DMSO-$d_6$, δ): 2.65 (3H, s, SO$_2$NHC$\underline{H}_3$), 3.76 (2H, t, J=5.6 Hz, SO$_2$CH$_2$), 3.86 (2H, q, J=5.2 Hz, CH$_2$O), 4.97 (1H, t, J=5.3 Hz, OH), 8.68 (1H, s, NH). $^{13}$C NMR (100 MHz, DMSO-$d_6$, δ): 28.31 (SO$_2$NH$\underline{C}$H$_3$), 55.07 (SO$_2\underline{C}$H$_2$), 59.54 ($\underline{C}$H$_2$O), 122.97 (C1, t, J ($^{19}$F – $^{13}$C)=15.0 Hz), 123.86 (C4, t, J ($^{19}$F – $^{13}$C)=15.7 Hz), 143.35 (C2 and C6, dd, $^1$J ($^{19}$F – $^{13}$C)=255.2 Hz, $^2$J ($^{19}$F – $^{13}$C)=15.7 Hz), 144.34 (C3 and C5, dd, $^1$J ($^{19}$F – $^{13}$C)=244.6 Hz, $^2$J ($^{19}$F – $^{13}$C)=13.3 Hz). $^{19}$F NMR (376 MHz, DMSO-$d_6$, δ): –136.00 (2F, dd, $^1$J=26.2 Hz, $^2$J=12.7 Hz), –136.61 to -136.77 (2F, m). HRMS for C$_9$H$_9$F$_4$NO$_5$S$_2$ [(M-H)$^-$]: calc. 349.9786, found 349.9785.

2-((2,3,5,6-Tetrafluoro-4-(N-methylsulfamoyl)phenyl)sulfonyl)ethyl acetate **(5bx)**. Yield: 0.035 g; (9%). Mp: 143 °C. $^1$H NMR (400 MHz, DMSO-$d_6$, δ): 1.85 (3H, s, COCH$_3$), 2.65 (3H, d, J=4.6 Hz, SO$_2$NHC$\underline{H}_3$), 4.01 (2H, t, J=5.4 Hz, SO$_2$CH$_2$), 4.41 (2H, t, J=5.7 Hz, CH$_2$O), 8.73 (1H, q, J=4.6 Hz, NH). $^{13}$C NMR (100 MHz, DMSO-$d_6$, δ): 20.11 (CO$\underline{C}$H$_3$), 28.32 (SO$_2$NHCH$_3$), 55.95 (SO$_2$CH$_2$), 57.21 (CH$_2$O), 121.74 (C1, t, J ($^{19}$F – $^{13}$C)=14.9 Hz), 124.46 (C4, t, J ($^{19}$F – $^{13}$C)=16.0 Hz), 143.64 (C2 and C6, d, $^1$J ($^{19}$F – $^{13}$C)=255.6 Hz), 144.57 (C3 and C5, d, $^1$J ($^{19}$F – $^{13}$C)=264.6 Hz), 169.56 (OC(O)). $^{19}$F NMR (376 MHz, DMSO-$d_6$, δ): –135.37 to -135.56 (2F, m), –136.09 to -136.30 (2 F, m). HRMS for C$_{11}$H$_{11}$F$_4$NO$_6$S$_2$ [(M-H)$^-$]: calc. 391.9891, found 391.9891.

2-((2,3,5,6-Tetrafluoro-4-sulfamoylphenyl)sulfonyl)ethyl acetate **(5ax)**. 2,3,5,6-tetrafluoro-4-((2-hydroxyethyl)sulfonyl)benzensulfonamide **(4ax)** (0.153 g; 0.45 mmol), acetic acid (0.260 ml; 4.54 mmol; 10 eq.) and one drop of H$_2$SO$_4$ (conc.) were dissolved in toluene (25 ml) and refluxed for 3 hours. The resulting mixture was cooled to 5 °C and white crystals were filtered. The product was purified by column chromatography (silica gel, EtOAc/CHCl$_3$ (1:1), Rf = 0.40). Yield: 0.134 g; (78%). Mp: 153–154°C. $^1$H NMR (400 MHz, DMSO-$d_6$, δ): 1.83 (3H, s, CH$_3$), 4.02 (2H, t, J=5.3 Hz, SO$_2$CH$_2$), 4.40 (2H, t, J=5.4 Hz, CH$_2$O), 8.69 (2H, s, SO$_2$NH$_2$). $^{13}$C NMR (100 MHz, DMSO-$d_6$, δ): 20.07 (CH$_3$), 55.91 (SO$_2$CH$_2$), 57.28 (CH$_2$O), 121.33 (C1, t, J ($^{19}$F – $^{13}$C)=15.3 Hz), 127.86 (C4, t, J ($^{19}$F – $^{13}$C)=15.7 Hz), 142.23 (C2 and C6, dd, $^1$J ($^{19}$F – $^{13}$C)=252.0 Hz, $^2$J ($^{19}$F – $^{13}$C)=20.2 Hz), 144.33 (C3 and C5, dd, $^1$J ($^{19}$F – $^{13}$C)=259.5 Hz, $^2$J ($^{19}$F – $^{13}$C)=18.6 Hz), 169.52 (OC(O)). $^{19}$F NMR (376 MHz, DMSO-$d_6$, δ): –135.60 to -135.78 (2F, m), –136.60 to -136.78 (2 F, m). HRMS for C$_{10}$H$_9$F$_4$NO$_6$S$_2$ [(M-H)$^-$]: calc. 377.9735, found 377.9735.

2,3,5,6-Tetrafluor-4-[2-(acetil)propilsulfonil]benzensulfonamidas **(5cx)**. 2,3,5,6-tetrafluoro-4-((3-hydroxypropyl)thio)benzenesulfonamide **(3cx)** (0.378 g; 0.95 mmol) was dissolved in acetic acid (10 ml) and heated at 75 °C temperature for 18 hr. H$_2$O$_2$ (30%) was added in portions (0.1 ml) every 30 min (overall 2.1 ml) until complete starting material conversion. Afterward, the solvent was evaporated under reduced pressure, and the product was purified by column chromatography (silica gel, EtOAc/CHCl$_3$ (1:1), Rf = 0.64). Yield: 0.216 g; (58%). Mp: 188 °C. $^1$H NMR (400 MHz, DMSO-$d_6$, δ): 2.00 (3H, s, CH$_3$), 2.03 (2H, p, J=7.8 Hz, CH$_2$C$\underline{H}_2$CH$_2$) 3.66 (2H, t, J=7.8 Hz CH$_2$SO$_2$), 4.08 (2H, t, J=6.4 Hz, CH$_2$O), 8.64 (2H, s, SO$_2$NH$_2$). $^{13}$C NMR (100 MHz, DMSO-$d_6$, δ): 20.66 (CH$_3$), 21.39 (CH$_2\underline{C}$H$_2$CH$_2$), 53.71 (SO$_2$CH$_2$), 61.58 (CH$_2$O), 120.30 (C1, t, J ($^{19}$F – $^{13}$C)=15.0 Hz), 127.74 (C4, t, J ($^{19}$F – $^{13}$C)=15.4 Hz), 142.99 (C2 and C6, d, J ($^{19}$F – $^{13}$C)=254.1 Hz), 144.46 (C3 and C5, d, J ($^{19}$F – $^{13}$C)=261.0 Hz), 170.39 (OC(O)). $^{19}$F NMR (376 MHz, DMSO-$d_6$, δ): –135,62–-135,82 (2F, m), –136,35–-136,58 (2 F, m). HRMS for C$_{11}$H$_{11}$F$_4$NO$_6$S$_2$ [(M-H)$^-$]: calc. 391.9891, found 391.891.

2-((2,3,5,6-tetrafluoro-4-sulfamoylphenyl)sulfonyl)ethyl propionate **(5dx)**. 2,3,5,6-tetrafluoro-4-((2-hydroxyethyl)sulfonyl)benzensulfonamide **(4ax)** (0.152 g; 0.451 mmol), propionic acid (2 ml) and three drops of H$_2$SO$_4$ (conc.) were dissolved in toluene (25 ml) and refluxed for 1 hr. The resulting mixture was washed with brine (3x10 ml). The organic phase was dried using anhydrous Na$_2$SO$_4$ and evaporated under reduced pressure. The product was purified by column chromotography (silica gel, EtOAc/CHCl$_3$ (1:1), Rf = 0.74). Yield: 0.0448 g; (25%). Mp: 109–111°C. $^1$H NMR (400 MHz, DMSO-$d_6$, δ): 0.93 (3H, t, J=7.5 Hz, CH$_2$C$\underline{H}_3$), 2.08 (2H, q, J=7.5 Hz, C$\underline{H}_2$CH$_3$), 4.03 (2H, t, J=5.6 Hz, SO$_2$CH$_2$), 4.42 (2H, t, J=5.6 Hz, CH$_2$O), 8.69 (2H, s, SO$_2$NH$_2$). $^{13}$C NMR (100 MHz, DMSO-$d_6$, δ): 8.63 (CH$_3$), 26.36 ($\underline{C}$H$_2$CH$_3$), 56.00 (SO$_2$CH$_2$), 57.22 (CH$_2$O), 121.32 (C1, t, J ($^{19}$F – $^{13}$C)=14.6 Hz), 127.85 (C4, t, J ($^{19}$F – $^{13}$C)=15.2 Hz), 142.90 (C2 and C6, dd, $^1$J ($^{19}$F – $^{13}$C)=256.1 Hz, $^2$J ($^{19}$F – $^{13}$C)=18.8 Hz), 144.36 (C2 and C5, dd, $^1$J ($^{19}$F – $^{13}$C)=259.1 Hz, $^2$J ($^{19}$F – $^{13}$C)=20.4 Hz), 172.82 (OC(O)). $^{19}$F NMR (376 MHz, DMSO-$d_6$, δ): –135.61 to -135.83 (2F, m), –136.58 to -136.79 (2 F, m). HRMS for C$_{11}$H$_{11}$F$_4$NO$_6$S$_2$ [(M-H)$^-$]: calc. 391.9891, found 391.9892.

2-((2,3,5,6-Tetrafluoro-4-sulfamoylphenyl)sulfonyl)ethyl pivalate **(5ex)**. 2,3,5,6-Tetrafluoro-4-((2-hydroxyethyl)sulfonyl)benzensulfonamide **(4ax)** (0.050 g; 0.015 mmol), pivalic acid (0.076 g; 0.074 mmol;

5 eq.) and three drops of $H_2SO_4$ (conc.) were dissolved in toluene (10 ml) and refluxed for 30 min. The resulting mixture was washed with brine (3x10 ml). The organic phase was dried using anhydrous $Na_2SO_4$ and evaporated under reduced pressure. The product was purified by column chromatography (silica gel, EtOAc/CHCl$_3$ (1:1), Rf = 0.62). Yield: 0.018 g; (29%). Mp: 67–68°C. $^1$H NMR (400 MHz, DMSO-d$_6$, δ): 1.02 (9H, s, C(CH$_3$)$_3$), 4.04 (2H, t, $J$=5.4 Hz, SO$_2$CH$_2$), 4.42 (2H, t, $J$=5.5 Hz, CH$_2$O), 8.70 (2H, s, SO$_2$NH$_2$). $^{13}$C NMR (100 MHz, DMSO-d$_6$, δ): 26.46 (C(CH$_3$)$_3$), 38.00 (C(CH$_3$)$_3$), 56.32 (SO$_2$CH$_2$), 57.35 (CH$_2$O), 121.12 (C1, t, $J$($^{19}$F – $^{13}$C)=14.9 Hz), 127.94 (C4, t, $J$($^{19}$F – $^{13}$C)=15.4 Hz), 143.01 (C2 and C6, dd, $^1J$($^{19}$F – $^{13}$C)=257.3 Hz, $^2J$($^{19}$F – $^{13}$C)=13.4 Hz), 144.35 (C3 and C5, dd, $^1J$($^{19}$F – $^{13}$C)=256.9 Hz, $^2J$($^{19}$F – $^{13}$C)=16.7 Hz), 176.84 (OC(O)). $^{19}$F NMR (376 MHz, DMSO-d$_6$, δ): –135.72 to -135.90 (2F, m), –136.46 to -136.63 (2 F, m). HRMS for $C_{13}H_{15}F_4NO_6S_2$ [(M-H)$^-$]: calc. 420.0204, found 420.0202.

2-((2,3,5,6-Tetrafluoro-4-sulfamoylphenyl)sulfonyl)ethyl 2-phenylacetate (**5fx**). 2,3,5,6-Tetrafluoro-4-((2-hydroxyethyl)sulfonyl)benzensulfonamide (**4ax**) (0.041 g; 0.122 mmol; 1 eq.), phenylacetic acid (0.075 g; 0.551 mmol; 5 eq.) and three drops of $H_2SO_4$ (conc.) were dissolved in toluene (15 ml) and refluxed for 1 hr. The resulting mixture was washed with brine (3x10 ml). The organic phase was dried using anhydrous $Na_2SO_4$ and evaporated under reduced pressure. The product was purified by column chromatography (silica gel, EtOAc/CHCl$_3$ (1:1), Rf = 0.71). Yield: 0.0282 g; (51%). Mp: 77–79°C. $^1$H NMR (400 MHz, DMSO-d$_6$, δ): 3.49 (2H, s, OC(O)CH$_2$), 4.05 (2H, t, $J$=5.5 Hz, SO$_2$CH$_2$), 4.46 (2H, t, $J$=5.5 Hz, CH$_2$O), 7.19 (2H, d, $J$=6.9 Hz, phenyl), 7.26 (1H, t, $J$=7.2 Hz, phenyl), 7,31 (2H, t, $J$=7.1 Hz, phenyl), 8.69 (2H, s, SO$_2$NH$_2$). $^{13}$C NMR (100 MHz, DMSO-d$_6$, δ): 40.68 (COCH$_2$), 55.96 (SO$_2$CH$_2$), 57.55 (CH$_2$O), 121.24 (C1, t, $J$($^{19}$F – $^{13}$C)=14.8 Hz), 126.95 (C4 of phenyl), 127.90 (C4, t, $J$($^{19}$F – $^{13}$C)=15.7 Hz), 128.36 (phenyl), 142.96 (C2 and C6, dd, $^1J$($^{19}$F – $^{13}$C)=259.0 Hz, $^2J$($^{19}$F – $^{13}$C)=17.7 Hz), 144.33 (C3 and C5, dd, $^1J$($^{19}$F – $^{13}$C)=254.2 Hz, $^2J$($^{19}$F – $^{13}$C)=18,6 Hz), 170.50 (OC(O)). $^{19}$F NMR (376 MHz, DMSO-d$_6$, δ): –135.53 to -135.70 (2F, m), –136.43 to -136.60 (2 F,m). HRMS for $C_{16}H_{13}F_4NO_6S_2$ [(M-H)$^-$]: calc. 454.0048, found 454.0044.

Methyl 3-((2,3,5,6-tetrafluoro-4-sulfamoylphenyl)thio)propanoate (**6x**). 2,3,4,5,6-Pentafluorbenzen sulfonamide (**2ax**) (0.675 g; 2.73 mmol), methyl 3-mercaptopropanoate (0.394 ml, 3.55 mmol, 1.3 eq.), Et$_3$N (0.456 ml, 3.28 mmol, 1.2 eq.) were dissolved in MeOH (15 ml) and left to stir at room temperature for 0.5 hr. After reaction completion, the solvent was evaporated under reduced pressure and the resultant precipitate was washed with $H_2O$. Recrystallization was accomplished from $H_2O$. Yield: 0.921 g; (97%). Mp: 124–125°C. $^1$H NMR (400 MHz, DMSO-d$_6$, δ): 2.66 (2H, t, $J$=6.8 Hz, CH$_2$C(O)), 3.22 (2H, t, $J$=6.8 Hz, SCH$_2$), 3.56 (3H, s, CH$_3$O), 8.43 (2H, s, SO$_2$NH$_2$). $^{13}$C NMR (100 MHz, DMSO-d$_6$, δ): 29.22 (SCH$_2$, t, $J$($^{19}$F – $^{13}$C)=2.8 Hz), 34.42 (CH$_2$C(O)O), 51.54 (CH$_3$O), 117.71 (C1, t, $J$($^{19}$F – $^{13}$C)=20.5 Hz), 122.86 (C4, t, $J$($^{19}$F – $^{13}$C)=15.6 Hz), 142.52 (C2 and C6, ddt, $^1J$($^{19}$F – $^{13}$C)=253.7 Hz, $^2J$($^{19}$F – $^{13}$C)=16.8 Hz, $^3J$($^{19}$F – $^{13}$C)=4.4 Hz), 146.70 (C3 and C5, ddt, $^1J$($^{19}$F – $^{13}$C)=243.6 Hz, $^2J$($^{19}$F – $^{13}$C)=15.8 Hz, $^3J$($^{19}$F – $^{13}$C)=3.4 Hz), 171.35 (C(O)O). $^{19}$F NMR (376 MHz, DMSO-d$_6$, δ): –132.67 to -132.83 (2F, m), –139.06 to -139.23 (2 F, m). HRMS for $C_{10}H_9F_4NO_4S_2$ [(M-H)$^-$]: calc. 345.9836, found 345.9836.

Methyl 3-((2,3,5,6-tetrafluoro-4-sulfamoylphenyl)sulfonyl)propanoate (**7x**). Methyl 3-((2,3,5,6-tetrafluoro-4-sulfamoylphenyl)thio)propanoate (**6x**) (0.043 g; 0.12 mmol) was dissolved in acetic acid (5 ml) and heated at 75 °C temperature for 4.5 hr. $H_2O_2$ (30%) was added in portions (0.1 ml) every 1.5 hr (overall 0.3 ml) until complete starting material conversion. Afterwards, the solvent was evaporated under reduced pressure and the product was recrystallized from $H_2O$. Yield: 0.032 g; (70%). Mp: 164 °C. $^1$H NMR (400 MHz, DMSO-d$_6$, δ): 2.84 (2H, t, $J$=7.2 Hz, CH$_2$C(O)), 3.59 (3H, s, CH$_3$O), 3.85 (2H, t, $J$=7.2 Hz, SO$_2$CH$_2$), 8.67 (2H, s, SO$_2$NH$_2$). $^{13}$C NMR (100 MHz, DMSO-d$_6$, δ): 26.93 (CH$_2$C(O)), 51.98 (CH$_3$O), 52.51 (SO$_2$CH$_2$), 120.14 (C1, t, $J$($^{19}$F – $^{13}$C)=15.0 Hz), 127.89 (C4, t, $J$($^{19}$F – $^{13}$C)=15.6 Hz), 143.01 (C2 and C6, dd, $^1J$($^{19}$F – $^{13}$C)=259.7 Hz, $^2J$($^{19}$F – $^{13}$C)=16.7 Hz), 144.56 (C3 and C5, dd, $^1J$($^{19}$F – $^{13}$C)=254.3 Hz, $^2J$($^{19}$F – $^{13}$C)=14.9 Hz), 170.12 (C(O)O). $^{19}$F NMR (376 MHz, DMSO-d$_6$, δ): –135,68 -: –135,88 (2F, m), –136,56 -: –136,76 (2F, m). HRMS for $C_{10}H_9F_4NO_6S_2$ [(M-H)$^-$]: calc. 377.9735, found 377.9735.

2-((2,3,5,6-Tetrafluoro-4-sulfamoylphenyl)thio)ethyl acetate (**8x**). 2,3,5,6-Tetrafluoro-4-((2-hydroxyethyl)thio)benzenesulfonamide (**3ax**) (0.104 g; 0.30 mmol; 1eq.), acetic acid (10 ml) and two drops of $H_2SO_4$ (conc.) were dissolved in toluene (35 ml) and refluxed for 2 hr. The resulting mixture was washed with brine (3x10 ml). The organic phase was dried using anhydrous $Na_2SO_4$ and evaporated under reduced pressure. The product was purified by column chromotography (silica gel, EtOAc/CHCl$_3$ (1:1), Rf = 0.76). Yield: 0.078 g; (65%). Mp: 108–110°C. $^1$H NMR (400 MHz, DMSO-d$_6$, δ): 1.90 (3H, s, CH$_3$),

3.29 (2H, t, $J$=5.9 Hz, SCH$_2$), 4.16 (2H, t, $J$=5.3 Hz, CH$_2$O), 8.44 (2H, s, SO$_2$NH$_2$). $^{13}$C NMR (100 MHz, DMSO-d$_6$, δ): 20.27 (CH$_3$), 32.56 (SCH$_2$, t, $J$ ($^{19}$F – $^{13}$C)=2.8 Hz), 63.15 (CH$_2$O), 117.89 (C4, t, $J$ ($^{19}$F – $^{13}$C)=20.5 Hz), 122.94 (C1, t, $J$ ($^{19}$F – $^{13}$C)=15.6 Hz), 142.48 (C3 and C5, ddt, $^1J$ ($^{19}$F – $^{13}$C)=253.7 Hz, $^2J$ ($^{19}$F – $^{13}$C)=17.2 Hz, $^3J$ ($^{19}$F – $^{13}$C)=4.4 Hz), 146.72 (C2 and C6, dd, $^1J$ ($^{19}$F – $^{13}$C)=240.8 Hz, $^2J$ ($^{19}$F – $^{13}$C)=13.9 Hz), 169.95 (OC(O)). $^{19}$F NMR (376 MHz, DMSO-d$_6$, δ): –132.44 to -132.69 (2F, m), –139.10 to -139.30 (2 F, m). HRMS for C$_{10}$H$_9$F$_4$NO$_4$S$_2$ [(M-H)$^-$]: calc. 345.9836, found 345.9838.

2,3,5,6-Tetrafluoro-4-((2-hydroxyethyl)sulfinyl)benzenesulfonamide (**9x**). 2,3,5,6-Tetrafluoro-4-((2-hydroxyethyl)thio)benzenesulfonamide (**3ax**) (0.095 g; 0.31 mmol; 1eq.), H$_2$O$_2$ (0.2 ml; 30%) were dissolved in acetic acid (4 ml) and left to stir at room temperature for 20 hr (after 2 hr additional portion of H$_2$O$_2$ (0.2 ml; 30%) was added). Afterward, the solvent was evaporated under reduced pressure and the product was purified by column chromatography (silica gel, EtOAc/CHCl$_3$ (1:1), Rf = 0.13). Yield: 0.081 g; (81%). Mp: 160–161°C. $^1$H NMR (400 MHz, DMSO-d$_6$, δ): 3.39–3.49 (1H, m, SOCH$_2$), 3.59 (1H, dt, $^1J$=13.2 Hz, $^2J$=4.0 Hz, SOCH$_2$), 3.84 (2H, q, $J$=4.8 Hz, CH$_2$OH), 5.18 (1H, t, $J$=5.0 Hz), 8.55 (2H, s, SO$_2$NH$_2$). $^{13}$C NMR (100 MHz, DMSO-d$_6$, δ): 54.20 (SO$_2$CH$_2$), 56.90 (CH$_2$O), 125.50 (C1, t, $J$=15.4 Hz), 126.09 (C4, t, $J$=17.6 Hz), 142.42 (C2 and C6, dd, $^1J$ ($^{19}$F – $^{13}$C)=247.8 Hz, $^2J$ ($^{19}$F – $^{13}$C)=16.4 Hz), 144.34 (C3 and C5, ddt, $^1J$ ($^{19}$F – $^{13}$C)=252.3 Hz, $^2J$ ($^{19}$F – $^{13}$C)=15.7 Hz, $^3J$ ($^{19}$F – $^{13}$C)=5.9 Hz). $^{19}$F NMR (376 MHz, DMSO-d$_6$, δ): –137.61 to -137.80 (2F, m), –139.07 to -139.26 (2 F, m). HRMS for C$_8$H$_7$F$_4$NO$_4$S$_2$ [(M+H)$^+$]: calc. 321.9825, found 321.9824.

2-((2,3,5,6-Tetrafluoro-4-sulfamoylphenyl)sulfinyl)ethyl acetate (**10x**). 2,3,5,6-Tetrafluoro-4-((2-hydroxyethyl)sulfinyl)benzenesulfonamide (**9x**) (0.081 g; 0.22 mmol), acetic acid (2 ml) and 3 drops of H$_2$SO$_4$ (conc.) were dissolved in toluene (30 ml) and refluxed for 1 hr. The resulting mixture was washed with brine (3x10 ml). The organic phase was dried using anhydrous Na$_2$SO$_4$ and evaporated under reduced pressure. The product was purified by column chromatography (silica gel, EtOAc/CHCl$_3$ (1:1), Rf = 0.36). Yield: 0.023 g; (25%). Mp: 132–133°C. $^1$H NMR (400 MHz, DMSO-d$_6$, δ): 1.96 (3H, s, CH$_3$), 3.68–3.81 (2H, m, SO$_2$CH$_2$), 4.36–4.49 (2H, m, CH$_2$O), 8.58 (2H, s, SO$_2$NH$_2$). $^{13}$C NMR (100 MHz, DMSO-d$_6$, δ): 20.30 (CH$_3$), 52.52 (SO$_2$CH$_2$), 57.06 (CH$_2$O), 125.33 (C1, t, $J$ ($^{19}$F – $^{13}$C)=17.1 Hz), 125.80 (C4, t, $J$ ($^{19}$F – $^{13}$C)=16.0 Hz), 142.47 (C2 and C6, dd, $^1J$ ($^{19}$F – $^{13}$C)=251.9 Hz, $^2J$ ($^{19}$F – $^{13}$C)=15.3 Hz), 144.35 (C3 and C5, $^1J$ ($^{19}$F – $^{13}$C)=252.7 Hz, $^2J$ ($^{19}$F – $^{13}$C)=14.9 Hz), 169.92 (OC(O)). $^{19}$F NMR (376 MHz, DMSO-d$_6$, δ): –137.51 to -137.68 (2F, m), –138.94 to -139.10 (2 F, m). HRMS for C$_{10}$H$_9$F$_4$NO$_5$S$_2$ [(M-H)$^-$]: calc. 361.9786, found 361.9794.

2,3,5,6-Tetrafluoro-4-(vinylsulfonyl)benzenesulfonamide (**11x**). 2,3,5,6-Tetrafluoro-4-((2-hydroxyethyl)sulfonyl)benzenesulfonamide (**3ax**) (0.200 g; 0.592 mmol; 1 eq.), thionyl chloride (0.064 ml; 0.85 mmol; 1.5 eq.), Et$_3$N (0.006 ml; 0.43 mmol; 0.75 eq.) were dissolved in MeCN (2 ml) and left to stir at room temperature for 52 hr. The resulting mixture was washed with brine (3x10 ml). The organic phase was dried using anhydrous Na$_2$SO$_4$ and evaporated under reduced pressure. The product was purified by column chromatography (silica gel, EtOAc/CHCl$_3$ (1:1), Rf = 0.64). Yield: 0.173 g; (92%). Mp: 174–176°C. $^1$H NMR (400 MHz, DMSO-d$_6$, δ): 6.54 (1H, d, $J$=17.0 Hz, CHCH$_2$), 6.57 (1H, d, $J$=23.6 Hz, CHCH$_2$), 7.36 (1H, dd, $^1J$=16.3 Hz, $^2J$=9.8 Hz, CHCH$_2$), 8.63 (2H, s, SO$_2$NH$_2$). $^{13}$C NMR (100 MHz, DMSO-d$_6$, δ): 121.79 (C1, t, $J$ ($^{19}$F – $^{13}$C)=14.2 Hz), 128.17 (C4, t, $J$ ($^{19}$F – $^{13}$C)=15.7 Hz), 133,34 (CH$_2$), 138,12 (CH), 143.45 (C3 and C5, dd, $^1J$ ($^{19}$F – $^{13}$C)=259.0 Hz, $^2J$ ($^{19}$F – $^{13}$C)=11.1 Hz) 144.49 (C2 and C6, dd, $^1J$ ($^{19}$F – $^{13}$C)=256.0 Hz, $^2J$ ($^{19}$F – $^{13}$C)=14.8 Hz). $^{19}$F NMR (376 MHz, DMSO-d$_6$, δ): –135.89 to -136.11 (2F, m), –136.43 to -136.72 (2 F, m). HRMS for C$_8$H$_5$F$_4$NO$_4$S$_2$ [(M-H)$^-$]: calc. 317.9523, found 317.9524.

2,3,5,6-Tetrafluoro-4-((1-hydroxy-2-methylpropan-2-yl)thio)benzenesulfonamide (**12x**). 2,3,4,5,6-pentafluorobenzensulfonamide (**2ax**) (0.753 g; 3.04 mmol; 1 eq.), 2-mercapto-2-methylpropan-1-ol (0.388 g; 3.66 mmol; 1.2 eq.), Et$_3$N (0.510 ml; 3.66 mmol; 1.2 eq.) were dissolved in MeOH (10 ml) and left to stir at room temperature overnight. Afterward, the solvent was evaporated under reduced pressure and the resultant precipitate was washed with H$_2$O. The product was purified by column chromatography (silica gel, EtOAc/CHCl$_3$ (1:1), Rf = 0.61). Yield: 0.690 g; (68%). Mp: 146–147°C. $^1$H NMR (400 MHz, DMSO-d$_6$, δ): 1.22 (6H, s, CH$_3$), 3.39 (2H, d, $J$=5.5 Hz, CH$_2$OH), 5.12 (1H, s, OH), 8.44 (2H, s, SO$_2$NH$_2$). $^{13}$C NMR (100 MHz, DMSO-d$_6$, δ): 25.29 (CH$_3$), 55.27 (CH$_2$OH), 69.96 (SC(CH$_3$)$_2$), 115.08 (C1, t, $J$ ($^{19}$F – $^{13}$C)=22.2 Hz), 124.57 (C4, t, $J$ ($^{19}$F – $^{13}$C)=15.6 Hz), 142.53 (C2 and C6, ddt, $^1J$ ($^{19}$F – $^{13}$C)=254.5 Hz, $^2J$ ($^{19}$F – $^{13}$C)=17.5 Hz, $^3J$ ($^{19}$F – $^{13}$C)=4.1 Hz), 148.32 (C3 and C5, dd, $^1J$ ($^{19}$F – $^{13}$C)=244.0 Hz, $^2J$ ($^{19}$F – $^{13}$C)=14.6 Hz). $^{19}$F NMR (376 MHz, DMSO-d$_6$, δ): –128.45 to -128.60 (2F, m), –138.67 to -138.84 (2 F, m). HRMS for C$_{10}$H$_{11}$F$_4$NO$_3$S$_2$ [(M+H)$^+$]: calc. 356.0009, found 355.9999.

Compounds **13x** and **14x** were synthesized during **12x** oxidation.

2,3,5,6-Tetrafluoro-4-((1-hydroxy-2-methylpropan-2-yl)thio)benzenesulfonamide (**12x**) (0.676 g, 2.03 mmol) was dissolved in acetic acid (10 ml) and heated at 75 °C temperature for 2.5 hr. $H_2O_2$ (30%) was added in portions (0.2 ml) every ~0.3 H (overall 0.8 ml) until complete starting material conversion. Afterward, the solvent was evaporated under reduced pressure and the product was purified by column chromatography (silica gel, EtOAc/CHCl$_3$ (1:1), Rf$_{14x\ (9)}$=0.66, Rf$_{13x}$=0.55).

2,3,5,6-Tetrafluoro-4-((1-hydroxy-2-methylpropan-2-yl)sulfonyl)benzenesulfonamide (**13x**). Yield: 0.145 g; (20%). Mp: 210–211°C. $^1$H NMR (400 MHz, DMSO-d$_6$, δ): 1.33 (6H, s, (CH$_3$)$_2$), 3.69 (2H, d, $J$=5.3 Hz, CCH$_2$), 5.22 (1H, t, $J$=4.2 Hz, OH), 8.61 (2H, s, SO$_2$NH$_2$). $^{13}$C NMR (100 MHz, DMSO-d$_6$, δ): 17.35 ((CH$_3$)$_2$), 63.92 (CH$_2$O), 67.74 (SO$_2$$\underline{C}$(CH$_3$)$_2$), 120.08 (C1, t, $J$ ($^{19}$F – $^{13}$C)=14.6 Hz), 127.79 (C4, t, $J$ ($^{19}$F – $^{13}$C)=15.7 Hz), 142.83 (C2 and C6, dd, $^1J$ ($^{19}$F – $^{13}$C)=252.0 Hz, $^2J$ ($^{19}$F – $^{13}$C)=15.7 Hz), 144.98 (C3 and C5, dd, $^1J$ ($^{19}$F – $^{13}$C)=257.4 Hz, $^2J$ ($^{19}$F – $^{13}$C)=17.7 Hz). $^{19}$F NMR (376 MHz, DMSO-d$_6$, δ): –132.27 to -132.44 (2F, m), –137.02 to -137.18 (2 F, m). HRMS for C$_{10}$H$_{11}$F$_4$NO$_5$S$_2$ [(M-H)$^+$]: calc. 366.0088, found 366.0088.

2-Methyl-2-((2,3,5,6-tetrafluoro-4-sulfamoylphenyl)sulfonyl)propyl acetate (**14x**). Yield: 0.153 g; (19%). Mp: 177–178°C. $^1$H NMR (400 MHz, DMSO-d$_6$, δ): 1.41 (6H, s, (CH$_3$)$_2$), 1.87 (3H, s, C(O)CH$_3$), 4.29 (2H, s, CCH$_2$), 8.64 (2H, s, SO$_2$NH$_2$). $^{13}$C NMR (100 MHz, DMSO-d$_6$, δ): 17.36 ((CH$_3$)$_2$), 19.92 (C(O)$\underline{C}$H$_3$), 65.31 (CH$_2$O), 65.76 (SO$_2$$\underline{C}$(CH$_3$)$_2$), 118.79 (C1, t, $J$ ($^{19}$F – $^{13}$C)=14.7 Hz), 128.33 (C4, t, $J$ ($^{19}$F – $^{13}$C)=15.8 Hz), 143.12 (C2 and C6, dd, $^1J$ ($^{19}$F – $^{13}$C)=258.8 Hz, $^2J$ ($^{19}$F – $^{13}$C)=13.9 Hz), 145.06 (C3 and C5, dd, $^1J$ ($^{19}$F – $^{13}$C)=255.9 Hz, $^2J$ ($^{19}$F – $^{13}$C)=13.9 Hz), 169.41 (OC(O)). $^{19}$F NMR (376 MHz, DMSO-d$_6$, δ): –132.52 to -132.70 (2F, m), –136.40 to -136.57 (2 F, m). HRMS for C$_{12}$H$_{13}$F$_4$NO$_6$S$_2$ [(M-H)$^-$]: calc. 406.0048, found 406.0052.

N-(2-((2,3,5,6-tetrafluoro-4-sulfamoylphenyl)thio)ethyl)acetamide (**15x**) was prepared according to known procedure in the literature (*Baronas et al., 2021*). 2,3,4,5,6-Pentafluorbenzensulfonamide (**2ax**) (2.320 g; 9.39 mmol), N-(2-mercaptoethyl)acetamide (1.300 ml; 13.1 mmol; 1.4 eq.), Et$_3$N (1.960 ml; 14.1 mmol; 1,5 eq.) were dissolved in MeOH (25 ml) and left to stir at room temperature for 2 hr. Afterward, additional portions of N-(2-mercaptoethyl)acetamide (0.150 ml; 1.51 mmol, 0.16 eq.) and Et$_3$N (0.150 ml; 1.08 mmol; 0.11 eq.) were added. The reaction mixture was left to stir further for an additional 1 hr. After reaction completion, the solvent was evaporated under reduced pressure and the product was purified by recrystallization from MeOH/H$_2$O 1:6 mixture obtaining white crystals. Yield 2.562 (78%). Mp: 169–170°C. (*Baronas et al., 2021*), mp: 169–171°C.

N-(2-((2,3,5,6-tetrafluoro-4-sulfamoylphenyl)sulfonyl)ethyl)acetamide (**16x**) was prepared as previously described (*Petrosiute et al., 2024*). N-(2-((2,3,5,6-tetrafluoro-4-sulfamoylphenyl)thio)ethyl)acetamide (**15x**) (2.592 g; 7.48 mmol; 1 eq.) was dissolved in acetic acid (70 ml) and heated at 75 °C temperature for 10 hr. $H_2O_2$ (30%) was added by portions (0.1 ml) every 30 min (overall 5 ml) until complete starting material conversion. Afterwards, the solvent was evaporated under reduced pressure and the product was purified by recrystallization in MeOH/H$_2$O 1:4 mixture obtaining white crystals. Yield: 1.576 g; (56%). Mp: 224–225°C, mp: 224–225°C.

N'-((4-bromophenyl)sulfonyl)-N,N-dimethylformimidamide (**18x**) was prepared according to the known procedure in literature (*Dudutienė et al., 2013*). N,N-Dimethylformamide dimethyl acetal (0.303 g; 2.54 mmol; 1.2 eq.) was dissolved in MeCN (7 ml) and added dropwise to a mixture of 4-bromobenzenesulfonamide (**17x**) (0.500 g; 2.12 mmol; 1 eq.) in MeCN (3 ml). The reaction mixture was left to stir at room temperature for 1 hr and afterward, the solvent was evaporated under reduced pressure and the resultant precipitate was washed with H$_2$O. Yield: 0.562 g; (91%). Mp: 142–143°C (close to the value in the literature [*Dudutienė et al., 2013*], mp: 141–143°C).

4-((2-Hydroxyethyl)thio)benzenesulfonamide (**19x**) was prepared according to known procedure in literature (*Dudutienė et al., 2013*). 2-Mercaptoethanol (0.161 g; 2.04 mmol; 2 eq.) was added dropwise to a suspension of NaH (0.089 g; 55% oil dispersion; 2.04 mmol; 2 eq.) in DMF (1 ml). After gas emission was complete, N'-((4-bromophenyl)sulfonyl)-N,N-dimethylformamidine (**18x**) (0.300 g; 1.02 mmol; 1eq.) was added and the reaction mixture was heated for 1 hr at 95 °C temperature. Subsequently, DMF was removed under reduced pressure, and the resultant precipitate was dissolved in a mixture of MeOH (1 ml) /NaOH solution (10%; 1 ml) and refluxed for another hour. MeOH was evaporated under reduced pressure and the resultant suspension was diluted with H$_2$O, washed with petrol ether, and acidified with HCl (10 %). Afterwards reaction mixture was extracted with EtOAc (3x10 ml), collected organic phase was dried using anhydrous Na$_2$SO$_4$ and evaporated under reduced

pressure. The product was purified by column chromatography (silica gel, EtOAc/CHCl$_3$ (1:1), Rf = 0.27). Yield 0.149 g; (63%). Mp: 110–111°C (close to the value in the literature [*Dudutienė et al., 2013*], mp: 111–112°C).

Compounds **20x** and **21x** (*Singh et al., 2012*) were synthesized during **19x** oxidation.

4-((2-Hydroxyethyl)thio)benzenesulfonamide (**19x**) (0.050 g, 0.21 mmol) was dissolved in acetic acid (2 ml) and heated at 75 °C temperature for 5 hr. H$_2$O$_2$ (30%) was added in portions (0.1 ml) every hour (overall 0.5 ml) until the complete starting material conversion. Afterward, the solvent was evaporated under reduced pressure and products were purified by column chromatography (silica gel, EtOAc/CHCl$_3$ (1:1), Rf$_{20x}$=0.09, Rf$_{21x\,(6)}$=0.34).

4-((2-Hydroxyethyl)sulfonyl)benzenesulfonamide **20x**. Yield: 0.032 g; (57%). Mp: 153–154°C. $^1$H NMR (400 MHz, DMSO-d$_6$, δ): 3.54 (2H, t, *J*=6.2 Hz SO$_2$CH$_2$), 3.70 (2H, t, *J*=6.0 Hz, CH$_2$O), 7.68 (2H, s, SO$_2$NH$_2$), 8.04 (2H, d, *J*=8.7 Hz, ArH), 8.10 (2H, d, *J*=8.8 Hz, ArH). $^{13}$C NMR (100 MHz, DMSO-d$_6$, δ): 54.98 (SO$_2$CH$_2$), 57.50 (CH$_2$O), 126.47 (C1), 128.65 (C4), 142.97 (C2 and C6), 148.40 (C3 and C5). HRMS for C$_8$H$_{11}$NO$_5$S$_2$ [(M-H)$^-$]: calc. 264.0006, found 264.0009.

2-((4-Sulfamoylphenyl)sulfonyl)ethyl acetate (**21x**). Yield: 0.004 g; (6%). Mp; 129–130°C. $^1$H NMR (400 MHz, DMSO-d$_6$, δ): 1.68 (3H, s, COCH$_3$), 3.82 (2H, t, *J*=5.6 Hz, SO$_2$CH$_2$), 4.27 (2H, t, *J*=5.6 Hz, CH$_2$O), 7.68 (2H, s, SO$_2$NH$_2$), 8.07 (2H, d, *J*=8.7 Hz, ArH), 8.12 (2H, d, *J*=8.7 Hz, ArH). $^{13}$C NMR (100 MHz, DMSO-d$_6$, δ): 20.10 (CH$_3$CO), 53.73 (SO$_2$CH$_2$), 57.49 (CH$_2$O), 126.57 (C1), 128.78 (C4), 142.49 (C2 and C6), 148.64 (C3 and C5), 169.59 (OC(O)). HRMS for C$_{15}$H$_{12}$F$_4$N$_2$O$_6$S$_2$ [(M-H)$^-$]: calc. 306.0112, found 306.0116.

N,N-dimethyl-N'-((2,3,5,6-tetrafluoro-4-((2-hydroxyethyl)sulfonyl)phenyl)sulfonyl)formamidine (**22x**). 2,3,5,6-Tetrafluoro-4-((2-hydroxyethyl)sulfonyl)benzensulfonamide (**4ax**) (0.160 g; 0.047 mmol) and N,N-dimethyl-formamide dimethyl acetal (0.076 g; 0.074 mmol; 5 eq.) were dissolved in acetonitrile (10 ml) and left to stir at room temperature for 1 hr. The solvent was evaporated under reduced pressure and the product was purified by column chromatography (silica gel, EtOAc, Rf = 0.65). Yield: 0.155 g; (83%). Mp: 188–189°C. $^1$H NMR (400 MHz, DMSO-d$_6$, δ): 2.99 (3H, s, NCH$_3$), 3.23 (3H, s, NCH$_3$), 3.71 (2H, t, *J*=5.3 Hz, SO$_2$CH$_2$), 3.85 (2H, q, *J*=5.3 Hz, CH$_2$O), 4.98 (1H, t, *J*=5.2 Hz, OH), 8.30 (1H, s, NCH). $^{13}$C NMR (100 MHz, DMSO-d$_6$, δ): 35.64 (NCH$_3$), 41.36 (NCH$_3$), 55.07 (SO$_2$CH$_2$), 59.53 (CH$_2$OH), 122.46 (C1, t, *J* ($^{19}$F – $^{13}$C)=15.1 Hz), 126.45 (C4, t, *J* ($^{19}$F – $^{13}$C)=15.2 Hz), 143.00 (C2 and C6, dd, $^1$*J* ($^{19}$F – $^{13}$C)=253.7 Hz, $^2$*J* ($^{19}$F – $^{13}$C)=17.8 Hz), 144.23 (C3 and C5, dd, $^1$*J* ($^{19}$F – $^{13}$C)=252.2 Hz, $^2$*J* ($^{19}$F – $^{13}$C)=16.3 Hz), 160.93 (NCHN). $^{19}$F NMR (376 MHz, DMSO-d$_6$, δ): −136.20 to -136.38 (2F, m), −136.72 to -136.91 (2 F, m). HRMS for C$_{11}$H$_{12}$F$_4$N$_2$O$_5$S$_2$ [(M+H)$^+$]: calc. 393.0197, found 393.0199.

2-((4-(N-((dimethylamino)methylene)sulfamoyl)–2,3,5,6-tetrafluorophenyl)sulfonyl)ethyl phenylcarbamate (**23ax**). N,N-dimethyl-N'-((2,3,5,6-tetrafluoro-4-((2-hydroxyethyl)sulfonyl) phenyl)sulfonyl) formamidine (**22x**) (0.068 g; 0.17 mmol; 1 eq.) and phenyl isocyanate (0.028 ml; 0.26 mmol; 1.5 eq.) were dissolved in toluene (10 ml) and refluxed at boiling point for 14 hr. Additional phenyl isocyanate portions (0.057 ml; 0.052 mmol; 3 eq.) were added after 2 hr and 7 hr, respectively. The solvent was evaporated under reduced pressure and the product was purified by column chromatography (silica gel, EtOAc/CHCl$_3$ (1:1), Rf = 0.43). Yield: 0.072 g; (81%). Mp: 177–178°C. $^1$H NMR (400 MHz, DMSO-d$_6$, δ): 2.96 (3H, s, NCH$_3$), 3.22 (3H, s, NCH$_3$), 4.04 (2H, t, *J*=5.3 Hz, SO$_2$CH$_2$), 4.51 (2H, t, *J*=5.4 Hz, CH$_2$O), 7.00 (1H, t, *J*=7.3 Hz, CH (C4) of phenyl), 7.27 (2H, t, *J*=7.8 Hz, CH (C3 and C5) of phenyl), 7.39 (2H, d, *J*=7.6 Hz, CH (C2 and C6) of phenyl), 8.27 (1H, s, NCHN), 9.62 (1H, s, OC(O)NH). $^{13}$C NMR (100 MHz, DMSO-d$_6$, δ): 35.61 (NCH$_3$), 41.34 (NCH$_3$), 56.36 (SO$_2$CH$_2$), 57.38 (CH$_2$O), 118.54 (C4 of phenyl), 121.10 (C1, t, *J* ($^{19}$F – $^{13}$C)=14.9 Hz), 122.75 (C3 and C5 of phenyl), 126.95 (C4, t, *J* ($^{19}$F – $^{13}$C)=15.1 Hz), 128.70 (C2 and C6 of phenyl), 138.54 (C1 of phenyl), 143.29 (C2 and C6, dd, $^1$*J* ($^{19}$F – $^{13}$C)=262.1 Hz, $^2$*J* ($^{19}$F – $^{13}$C)=22.0 Hz), 144.25 (C5 and C3, dd, $^1$*J* ($^{19}$F – $^{13}$C)=247.2 Hz, $^2$*J* ($^{19}$F – $^{13}$C)=13.0 Hz), 152.64 (OC(O)NH), 160.89 (NCHN). $^{19}$F NMR (376 MHz, DMSO-d$_6$, δ): −135.99 to -136.16 (2F, m), −136.28 to -136.47 (2 F, m). HRMS for C$_{18}$H$_{17}$F$_4$N$_3$O$_6$S$_2$ [(M+H)$^+$]: calc. 512.0568, found 512.0571.

2-((4-(N-((dimethylamino)methylene)sulfamoyl)–2,3,5,6-tetrafluorophenyl)sulfonyl)ethyl (4-methoxyphenyl)carbamate (**23bx**). N,N-dimethyl-N'-((2,3,5,6-tetrafluoro-4-((2-hydroxyethyl) sulfonyl) phenyl)sulfonyl)formamidine (**22x**) (0.050 g; 0.13 mmol; 1 eq.), 4-methoxyphenyl isocyanate (0.025 ml; 0.19 mmol; 1.5 eq.), dibutyltin dilaurate (0.015 ml; 0.026 mmol; 0.2 eq.) were dissolved in acetonitrile (3 ml) and left to stir at room temperature for 5 d. An additional 4-methoxyphenyl isocyanate portion (0.025 ml; 0.19 mmol; 1.5 eq.) was added after 2 d. The solvent was evaporated under reduced pressure and the product was purified by column chromatography (silica gel, EtOAc/

CHCl$_3$ (1:1), Rf = 0.41). Yield: 0.008 g; (12%). Mp: 167–168°C. $^1$H NMR (400 MHz, DMSO-d$_6$, δ): 2.96 (3H, s, NCH$_3$), 3.21 (3H, s, NCH$_3$), 3.70 (3H, s, OCH$_3$), 4.03 (2H, t, J=5.4 Hz, SO$_2$CH$_2$), 4.49 (2H, t, J=5.3 Hz, CH$_2$O), 6.85 (2H, d, J=8.8 Hz, CH (C2 and C6) of phenyl), 7.28 (2H, d, J=7.2 Hz, CH (C3 and C6) of phenyl), 8.27 (1H, s, NCHN), 9.41 (1H, s, OC(O)NH). $^{13}$C NMR (100 MHz, DMSO-d$_6$, δ): 35.58 (NCH$_3$), 41.34 (NCH$_3$), 55.14 (OCH$_3$), 56.40 (SO$_2$CH$_2$), 57.31 (CH$_2$O), 113.91 (C3 and C5 of phenyl), 120.25 (C2 and C6 of phenyl), 121.13 (C1, t, J ($^{19}$F – $^{13}$C)=14.7 Hz), 126.91 (C4), 128.63 (C1 of phenyl), 143.34 (C2 and C6, dd, $^1$J ($^{19}$F – $^{13}$C)=250.0 Hz, $^2$J ($^{19}$F – $^{13}$C)=11.2 Hz), 144.16 (C3 and C5, dd, $^1$J ($^{19}$F – $^{13}$C)=242.7 Hz, $^2$J ($^{19}$F – $^{13}$C)=10.3 Hz), 152.79 (OC(O)NH), 155.03 (C4 of phenyl), 160.90 (NCHN). $^{19}$F NMR (376 MHz, DMSO-d$_6$, δ): −135.95 to -136.16 (2F, m), −136.30 to -136.55 (2 F,m). HRMS for C$_{19}$H$_{19}$F$_4$N$_3$O$_7$S$_2$ [(M+Na)$^+$]: calc. 564.0493, found 564.0501.

2-((2,3,5,6-Tetrafluoro-4-sulfamoylphenyl)sulfonyl)ethyl phenylcarbamate (**24ax**) (*Figure 11*). 2-((4-(N-((dimethylamino)methylene)sulfamoyl)–2,3,5,6-tetrafluorophenyl)sulfonyl)ethyl phenylcarbamate (**23ax**) (0.030 g; 0.059 mmol; 1eq.) and three drops of HCl (conc.) were dissolved in toluene (3 ml) and refluxed at boiling point for 18 hr. After 5 hr, additional 10 drops of HCl (conc.) were added. The solvent was evaporated under reduced pressure and the product was purified by column chromatography (silica gel, EtOAc/CHCl$_3$ (1:1), Rf = 0.59). Yield: 0.007 g; (26%). Mp: 204–205°C. $^1$H NMR (400 MHz, DMSO-d$_6$, δ): 4.06 (2H, t, J=5.4 Hz, SO$_2$CH$_2$), 4.52 (2H, t, J=5.5 Hz, CH$_2$O), 7.00 (1H, t, J=7.3 Hz, CH (C4) of phenyl), 7.26 (2H, t, J=7.9 Hz, (C3 and C5) of phenyl), 7.41 (2H, d, J=7.9 Hz, CH (C2 and C6) of phenyl), 8.60 (2H, s, SO$_2$NH$_2$), 9.65 (1H, s, OC(O)NH). $^{13}$C NMR (100 MHz, DMSO-d$_6$, δ): 56.39 (SO$_2$CH$_2$), 57.26 (CH$_2$O), 118.51 (C4 of phenyl), 121.18 (C1, t, J ($^{19}$F – $^{13}$C)=13.1 Hz), 122.72 (C3 and C5 of phenyl), 127.86 (C4, t, J ($^{19}$F – $^{13}$C)=15.0 Hz), 128.71 (C2 and C6 of phenyl), 138.57 (C1 of phenyl), 143.12 (C2 and C6, dd, $^1$J ($^{19}$F – $^{13}$C)=257.4 Hz, $^2$J ($^{19}$F – $^{13}$C)=17.7 Hz), 144.26 (C3 and C5, dd, $^1$J ($^{19}$F – $^{13}$C)=259.2 Hz, $^2$J ($^{19}$F – $^{13}$C)=19.5 Hz), 152.65 (OC(O)NH). $^{19}$F NMR (376 MHz, DMSO-d$_6$, δ): −136.18 to -136.49 (4F, m). HRMS for C$_{15}$H$_{12}$F$_4$N$_2$O$_6$S$_2$ [(M-H)$^-$]: calc. 455.0000, found 455.0001.

2-((2,3,5,6-Tetrafluoro-4-sulfamoylphenyl)sulfonyl)ethyl (4-methoxyphenyl)carbamate (**24bx**). 2-((4-(N-((dimethylamino)methylene)sulfamoyl)–2,3,5,6-tetrafluorophenyl)sulfonyl)ethyl (4-methoxyphenyl)carbamate (**23bx**) (0.006 g; 0.011 mmol; 1 eq.) and three drops of HCl (conc.) were dissolved in MeOH (3 ml) and refluxed for 31 hr. The solvent was evaporated under reduced pressure and the product was purified by column chromatography (silica gel, EtOAc/CHCl$_3$, (1:1), Rf = 0.57). Yield: 0.002 g; (37%). Mp: 188–189°C. $^1$H NMR (400 MHz, DMSO-d$_6$, δ): 3.70 (3H, s, OCH$_3$), 4.05 (2H, t, J=5.4 Hz, SO$_2$CH$_2$), 4.49 (2H, t, J=5.4 Hz, CH$_2$O), 6.85 (2H, d, J=8.9 Hz, phenyl), 7.30 (2H, d, J=7.5 Hz, phenyl), 8.61 (2H, s, SO$_2$NH$_2$), 9.45 (1H, s, OC(O)NH). $^{13}$C NMR (100 MHz, DMSO-d$_6$, δ): 55.16 (OCH$_3$), 56.44 (SO$_2$CH$_2$), 57.23 (CH$_2$O), 113.94 (C3 and C5 of phenyl), 120.14 (C2 and C6 of phenyl), 121.23 (C1), 127.85 (C4, t, J ($^{19}$F – $^{13}$C)=12.2 Hz), 143.06 (C2 and C6, d, J ($^{19}$F – $^{13}$C)=255.9 Hz), 144.26 (C3 and C5, dd, $^1$J ($^{19}$F – $^{13}$C)=259.6 Hz, $^2$J ($^{19}$F – $^{13}$C)=19.2 Hz), 152.82 (OC(O)NH), 155.01 (C4 of phenyl). $^{19}$F NMR (376 MHz, DMSO-d$_6$, δ): −136.22 to -136.49 (4F, m). HRMS for C$_{16}$H$_{14}$F$_4$N$_2$O$_7$S$_2$ [(M-H)$^-$]: calc. 485.0106, found 485.0105.

3-(Cyclooctylamino)–2,5,6-trifluoro-4-((2-hydroxyethyl)sulfonyl)benzenesulfonamide (**25ax**) was prepared according to the known procedure in literature (*Dudutienė et al., 2014*). 2,3,5,6-Tetrafluoro-4-((2-hydroxyethyl)sulfonyl)benzensulfonamide (**4ax**) (0.549 g; 1.63 mmol; 1 eq.) and cyclooctylamine (0.446 ml; 3.26 mmol; 2 eq.) were dissolved in DMSO (2 ml) and left to stir at room temperature overnight. After full starting material conversion reaction mixture was washed with brine (10 ml) and extracted with EtOAc (3x15 ml). The organic phase was dried using anhydrous Na$_2$SO$_4$ and evaporated under reduced pressure. The product was purified by column chromatography (silica gel, EtOAc/CHCl$_3$ (1:1), Rf = 0.52). Yield 0.380 g; (52%). Mp: 90–91°C (close to the value in the literature [*Dudutienė et al., 2014*], mp: 89–90°C).

3-(Cyclododecylamino)–2,5,6-trifluoro-4-((2-hydroxyethyl)sulfonyl)benzenesulfonamide (**25bx**) was prepared according to known procedure in literature (*Dudutienė et al., 2015*). 2,3,5,6-Tetrafluoro-4-((2-hydroxyethyl)sulfonyl)benzensulfonamide (**4ax**) (0.300 g 0.89 mmol; 1 eq.) and cyclododecylamine (0.326 g; 1.78 mmol; 2 eq.) were dissolved in DMSO (2 ml) and left to stir at room temperature for 2 hrs. After full starting material conversion reaction mixture was washed with brine (10 ml) and extracted with EtOAc (3x15 ml). The organic phase was dried using anhydrous Na$_2$SO$_4$ and evaporated under reduced pressure. The product was purified by column chromatography (silica gel, EtOAc/CHCl$_3$ (1:1), Rf = 0.60). Yield: 0.278 g; (62%). Mp: 143–145°C (close to the value in the literature [*Dudutienė et al., 2015*], mp: 143–144°C).

N'-((3-(Cyclooctylamino)–2,5,6-trifluoro-4-((2-hydroxyethyl)sulfonyl)phenyl)sulfonyl)-N,N-dimethylformamidine (**26x**). N,N-dimethyl-N'-((2,3,5,6-tetrafluoro-4-((2-hydroxyethyl)sulfonyl) phenyl) sulfonyl)formamidine (**22x**) (0.100 g; 0.25 mmol; 1 eq.) and cyclooctylamine (0.070 ml; 0.51 mmol; 2 eq.) were dissolved in DMSO (1 ml) and left to stir at room temperature for 1.5 hr. The reaction mixture was washed with brine (5 ml) and extracted using EtOAc (3x10 ml). The collected organic phase was dried using anhydrous $Na_2SO_4$ and evaporated under reduced pressure. The product was purified by column chromatography (silica gel, EtOAc, Rf = 0.64) and yellow oil was obtained. Yield: 0.062 g; (49%). $^1$H NMR (400 MHz, DMSO-$d_6$, δ): 1.40–1.70 (12H, m, cyclooctane), 1.76–1.87 (2H, m, cyclooctane), 2.97 (3H, s, $NCH_3$), 3.21 (3H, s, $NCH_3$), 3.64 (2H, t, J=5.4 Hz, $SO_2CH_2$), 3.73 (1H, br.s, NHC$\underline{H}$(CH$_2$)$_2$), 3.81 (2H, q, J=5,4 Hz, $CH_2O$), 4.98 (1H, t, J=5.2 Hz, OH), 6.60 (1H, d, J=8.2 Hz, N$\underline{H}$CH(CH$_2$)$_2$), 8.28 (1H, s, NCHN). $^{13}$C NMR (100 MHz, DMSO-$d_6$, δ): 22.89 (cyclooctane), 25.11 (cyclooctane), 26.65 (cyclooctane), 32.27 (cyclooctane), 35.48 ($NCH_3$), 41.20 ($NCH_3$), 55.02 ($CH_2O$), 55.31 (CH of cyclooctane, d, J ($^{19}$F – $^{13}$C)=11.1 Hz), 59.52 ($SO_2CH_2$, d, J ($^{19}$F – $^{13}$C)=2.5 Hz), 116.70 (C1, dd, $^1$J ($^{19}$F – $^{13}$C)=12.8 Hz, $^2$J ($^{19}$F – $^{13}$C)=5.5 Hz), 126.05 (C4, dd, $^1$J ($^{19}$F – $^{13}$C)=18.2 Hz, $^2$J ($^{19}$F – $^{13}$C)=13.9 Hz), 134.49 (C3, d, J ($^{19}$F – $^{13}$C)=16.1 Hz), 136.97 (C6, d, J ($^{19}$F – $^{13}$C)=246.7 Hz), 144.26 (C2, d, $^1$J ($^{19}$F – $^{13}$C)=252.5 Hz), 145.62 (C5, dd, $^1$J ($^{19}$F – $^{13}$C)=246.0 Hz, $^1$J ($^{19}$F – $^{13}$C)=12.4 Hz), 160.75 (NCHN). $^{19}$F NMR (376 MHz, DMSO-$d_6$, δ): –124.67 (1F, br.s), –134.26 (1F, dd, $J^1$=27.3 Hz, $J^2$ = 12.2 Hz), –150.64 (1F, dd, $J^1$=27.3 Hz, $J^2$ = 6.7 Hz). HRMS for $C_{19}H_{28}F_3N_3O_5S_2$ [(M+H)$^+$]: calc. 500.1495, found 500.1493.

2-((2-(Cyclooctylamino)–3,5,6-trifluoro-4-sulfamoylphenyl)sulfonyl)ethyl acetate (**27ax**). 3-(Cyclooctylamino)–2,5,6-trifluoro-4-((2-hydroxyethyl)sulfonyl)benzenesulfonamide (**25ax**) (0.040 g; 0.090 mmol), acetic acid (1.2 ml) and two drops of $H_2SO_4$ (conc.) were dissolved in toluene (10 ml) and refluxed for 1 hr. The resulting mixture was washed with brine (3x10 ml). The organic phase was dried using anhydrous $Na_2SO_4$ and evaporated under reduced pressure. The product was purified by column chromotography (silica gel, EtOAc/CHCl$_3$ (1:1), Rf = 0.78). Yield: 0.0203 g; (46%). Mp: 96 °C. $^1$H NMR (400 MHz, CDCl$_3$, δ): 1.45–1.73 (12H, m, cyclooctane), 1.83–1.91 (2H, m, cyclooctane), 1.93 (3H, s, $CH_3$), 3.66 (2H, t, J=5.7 Hz, $SO_2CH_2$), 3.88 (1H, s, cyclooctane), 4.48 (2H, t, J=5.7 Hz, $CH_2O$), 5.65 (2H, s, $SO_2NH_2$), 6.84 (2H, d, J=6.7 Hz, NH). $^{13}$C NMR (100 MHz, CDCl$_3$, δ): 20.46 ($CH_3$), 23.38 (cyclooctane), 25.54 (cyclooctane), 27.31 (cyclooctane), 33.04 (cyclooctane), 56.24 (cyclooctane, d, J ($^{19}$F-$^{13}$C)=11.7 Hz), 56.43 ($SO_2CH_2$, d, J ($^{19}$F-$^{13}$C)=4.0 Hz), 57.39 ($CH_2O$), 115.10 (C1, dd, $^1$J ($^{19}$F-$^{13}$C)=12.7 Hz, $^2$J ($^{19}$F-$^{13}$C)=5.9 Hz), 126.63 (C4, dd, $^1$J ($^{19}$F-$^{13}$C)=17.1 Hz, $^2$J ($^{19}$F-$^{13}$C)=13.3 Hz), 135.85 (C3, d, J ($^{19}$F-$^{13}$C)=13.6 Hz), 136.57 (C6, ddd, $^1$J ($^{19}$F-$^{13}$C)=248.33 Hz, $^2$J ($^{19}$F-$^{13}$C)=17.8 Hz, $^3$J ($^{19}$F-$^{13}$C)=3.9 Hz), 144.36 (C2, d, $^1$J ($^{19}$F-$^{13}$C)=252.7 Hz), 146.28 (C5, dd, $^1$J ($^{19}$F-$^{13}$C)=252.8 Hz, $^2$J ($^{19}$F-$^{13}$C)=15.8 Hz, $^3$J ($^{19}$F-$^{13}$C)=4.5 Hz), 170.29 (OC(O)). $^{19}$F NMR (376 MHz, CDCl$_3$, δ): –125.63 (1F, dd, $^1$J=12.4 Hz, $^2$J=8.8 Hz), –132.86 (1F, dd, $^1$J=25.7 Hz, $^2$J=12.5 Hz), –151.63 (1F, dd, $^1$J=25.8 Hz, $^2$J=8.8 Hz). HRMS for $C_{18}H_{25}F_3N_2O_6S_2$ [(M+H)$^+$]: calc. 487.1179, found 487.1179.

2-((2-(Cyclododecylamino)–3,5,6-trifluoro-4-sulfamoylphenyl)sulfonyl)ethyl acetate (**27bx**). 3-(Cyclododecylamino)–2,5,6-trifluoro-4-((2-hydroxyethyl)sulfonyl)benzenesulfonamide (**25bx**) (0.102 g; 0.204 mmol), acetic acid (4 ml) and three drops of $H_2SO_4$ (conc.) were dissolved in toluene (25 ml) and refluxed for 4 hr. The resulting mixture was washed with brine (3x10 ml). The organic phase was dried using anhydrous $Na_2SO_4$ and evaporated under reduced pressure. The product was purified by column chromatography (silica gel, EtOAc/CHCl$_3$ (1:1), Rf = 0.86). Yield: 0.052 g; (47%). Mp: 138–140°C. $^1$H NMR (400 MHz, DMSO-$d_6$, δ): 1.21–1.46 (20H, m, cyclododecane), 1.55–1.65 (2H, m, cyclododecane), 1.81 (3H, s, OC(O)CH$_3$), 3.78 (1H, s, cyclododecane), 3.92 (2H, t, J=4.9 Hz, $SO_2CH_2$), 4.36 (2H, t, J=4.9 Hz, $CH_2O$), 6.52 (1H, d, J=8.6 Hz, NH), 8.38 (2H, s, $SO_2NH_2$). $^{13}$C NMR (100 MHz, DMSO-$d_6$, δ): 20.04 ($CH_3$), 20.45 (cyclododecane), 22.60 (cyclododecane), 22.71 (cyclododecane), 23.73 (cyclododecane), 23.89 (cyclododecane), 30.08 (cyclododecane), 52.84 (CH of cyclododecane, d, J=11.6 Hz), 55.66 ($SO_2CH_2$, d, J=2.3 Hz), 57.65 ($CH_2O$), 115.34 (C1, dd, $^1$J ($^{19}$F – $^{13}$C)=13.2 Hz, $^2$J ($^{19}$F – $^{13}$C)=5.1 Hz), 127.71 (C4, dd, $^1$J ($^{19}$F – $^{13}$C)=18.1 Hz, $^2$J ($^{19}$F – $^{13}$C)=13.9 Hz), 135.13 (C3, d, $^1$J ($^{19}$F – $^{13}$C)=15.3 Hz), 136.67 (C6, d, $^1$J ($^{19}$F – $^{13}$C)=253.6 Hz), 143.95 (C2, d, $^1$J ($^{19}$F – $^{13}$C)=252.9 Hz), 145.58 (C5, dd, $^1$J ($^{19}$F – $^{13}$C)=246.4 Hz, $^2$J ($^{19}$F – $^{13}$C)=13.7 Hz), 169.54 (OC(O)). $^{19}$F NMR (376 MHz, DMSO-$d_6$, δ): –124.88 (1F, s), –134.22 (1F, dd, $^1$J=26.8 Hz, $^2$J=12.5 Hz), –150.76 (1F, dd, $^1$J=26.8 Hz, $^2$J=6.7 Hz). HRMS for $C_{22}H_{33}F_3N_2O_6S_2$ [(M+H)$^+$]: calc. 543.1805, found 543.1803.

2-((2-(Cyclooctylamino)–3,5,6-trifluoro-4-sulfamoylphenyl)sulfonyl)ethyl propionate (**28ax**). 3-(Cyclooctylamino)–2,5,6-trifluoro-4-((2-hydroxyethyl)sulfonyl)benzenesulfonamide (**25ax**) (0.100 g; 0.225 mmol), propionic acid (3 ml) and three drops of $H_2SO_4$ (conc.) were dissolved in toluene (25 ml)

and refluxed for 3 hr. The resulting mixture was washed with brine (3x10 ml). The organic phase was dried using anhydrous $Na_2SO_4$ and evaporated under reduced pressure. The product was purified by column chromatography (silica gel, EtOAc/CHCl$_3$ (1:1), Rf = 0.89) and yellow oil was obtained. Yield: 0.0313 g; (28%). $^1$H NMR (400 MHz, CDCl$_3$, δ): 1.06 (3H, t, J=7.6 Hz, CH$_2$CH$_3$), 1.45–1.74 (12H, m, cyclooctane), 1.83–1.92 (2H, m, cyclooctane), 2.17 (2H, q, J=7.6 Hz, CH$_2$CH$_3$), 3.66 (2H, t, J=5.8 Hz, SO$_2$CH$_2$), 3.88 (1H, s, CH of cyclooctane), 4.49 (2H, t, J=5.8 Hz, CH$_2$O), 5.55 (2H, s, SO$_2$NH$_2$), 6,85 (1H, d, J=8.0 Hz, NH). $^{13}$C NMR (100 MHz, CDCl$_3$, δ):): 8.88 (CH$_2$CH$_3$), 23.40 (cyclooctane), 25.56 (cyclooctane), 27.19 (CH$_2$CH$_3$), 27.31 (cyclooctane), 33.06 (cyclooctane), 56.25 (CH of cyclooctane, d, J ($^{19}$F-$^{13}$C)=11.7 Hz), 56.58 (SO$_2$CH$_2$, d, J ($^{19}$F-$^{13}$C)=3.9 Hz), 57.28 (CH$_2$O), 115.18 (C1, dd, $^1$J ($^{19}$F-$^{13}$C)=13.3 Hz, $^2$J ($^{19}$F-$^{13}$C)=6.5 Hz), 126.63 (C4, dd, $^1$J ($^{19}$F-$^{13}$C)=16.8 Hz, $^2$J ($^{19}$F-$^{13}$C)=13.4 Hz), 135.89 (C3, d, J ($^{19}$F-$^{13}$C)=12.9 Hz), 136.59 (C6, ddd, $^1$J ($^{19}$F-$^{13}$C)=248.3 Hz, $^2$J ($^{19}$F-$^{13}$C)=17.4 Hz, $^3$J ($^{19}$F-$^{13}$C)=4.3 Hz), 144.39 (C2, d, $^1$J ($^{19}$F-$^{13}$C)=253.3 Hz, $^2$J ($^{19}$F-$^{13}$C)=3.7 Hz), 146.30 (C5, ddd, $^1$J ($^{19}$F-$^{13}$C)=253.2 Hz, $^2$J ($^{19}$F-$^{13}$C)=16.5 Hz, $^3$J ($^{19}$F-$^{13}$C)=5.0 Hz), 173.73 (OC(O)). $^{19}$F NMR (376 MHz, CDCl$_3$, δ): –125.61 (1F, dd, $^1$J=12.4 Hz, $^2$J=8.9 Hz), –132.79 (1F, dd, $^1$J=25.8 Hz, $^2$J=12.5 Hz), –151.60 (1F, dd, $^1$J=25.8 Hz, $^2$J=8.8 Hz). HRMS for C$_{19}$H$_{27}$F$_3$N$_2$O$_6$S$_2$ [(M+H)$^+$]: calc. 501.1335, found 501.1339.

2-((2-(Cyclododecylamino)–3,5,6-trifluoro-4-sulfamoylphenyl)sulfonyl)ethyl propionate (**28bx**). 3-(Cyclododecylamino)–2,5,6-trifluoro-4-((2-hydroxyethyl)sulfonyl)benzenesulfonamide (**25bx**) (0.0377 g; 0.075 mmol), propionic acid (2 ml) and three drops of H$_2$SO$_4$ (conc.) were dissolved in toluene (10 ml) and refluxed for 1 hr. The resulting mixture was washed with brine (2x5 ml). The organic phase was dried using anhydrous Na$_2$SO$_4$ and evaporated under reduced pressure. The product was purified by column chromatography (silica gel, CHCl$_3$, Rf = 0.20). Yield: 0.0248 g; (59%). Mp: 110–112°C. $^1$H NMR (400 MHz, DMSO-d$_6$, δ): 0.91 (3H, t, J=7.5 Hz, CH$_2$CH$_3$), 1.23–1.44 (20H, m, cyclododecane), 1.54–1.64 (2H, m, cyclododecane), 2.05 (2H, q, J=7.6 Hz, CH$_2$CH$_3$), 3.78 (1H, s, cyclododecane), 3.94 (2H, t, J=5.2 Hz, SO$_2$CH$_2$), 4.37 (2H, t, J=5.0 Hz, CH$_2$O), 6.52 (1H, d, J=8.2 Hz, NH), 8.40 (2H, s, SO$_2$NH$_2$). $^{13}$C NMR (100 MHz, DMSO-d$_6$, δ): 8.56 (CH$_2$CH$_3$), 20.44 (cyclododecane), 22.58 (cyclododecane), 22.70 (cyclododecane), 23.71 (cyclododecane), 23.87 (cyclododecane), 26.36 (CH$_2$CH$_3$), 30.08 (cyclododecane), 52.82 (CH of cyclododecane, d, J=11.6 Hz), 55.76 (SO$_2$CH$_2$), 57.59 (CH$_2$O), 115.32 (C1, t, J ($^{19}$F – $^{13}$C)=7.0 Hz), 127.67 (C4, t, J ($^{19}$F – $^{13}$C)=14.2 Hz), 135.11 (dd, $^1$J ($^{19}$F – $^{13}$C)=13.9 Hz, $^2$J ($^{19}$F – $^{13}$C)=2.4 Hz), 136.65 (C6, d, $^1$J ($^{19}$F – $^{13}$C)=252.8 Hz), 143.98 (C2, d, $^1$J ($^{19}$F – $^{13}$C) 253.0 Hz,), 145.56 (C5, dd, $^1$J ($^{19}$F – $^{13}$C)=249.8 Hz, $^2$J ($^{19}$F – $^{13}$C)=15.4 Hz), 172.82 (OC(O)). $^{19}$F NMR (376 MHz, DMSO-d$_6$, δ): –124.86 (1F, s), –134.21 (1F, dd, $^1$J=26.9 Hz, $^2$J=12.4 Hz), –150.79 (1F, dd, $^1$J=26.9 Hz, $^2$J=6.6 Hz). HRMS for C$_{23}$H$_{35}$F$_3$N$_2$O$_6$S$_2$ [(M+H)$^+$]: calc. 557.1961, found 557.1966.

2-((2-(Cyclooctylamino)–4-(N-((dimethylamino)methylene)sulfamoyl)–3,5,6-trifluorophenyl) sulfonyl) ethyl phenylcarbamate (**29x**). N'-((3-(cyclooctylamino)–2,5,6-trifluoro-4-((2-hydroxyethyl)sulfonyl) phenyl)sulfonyl)-N,N-dimethylformamidine (**26x**) (0.038 g; 0.08 mmol; 1 eq.) and phenyl isocyanate (0.025 ml; 0.23 mmol; 3 eq.) were dissolved in toluene (7 ml) and refluxed at boiling point for 28 hr. Additional phenyl isocyanate portions (0.025 ml; 0.023 mmol; 3 eq.) were added after 10 hours and 24 hours respectively. The solvent was evaporated under reduced pressure and the product was purified by column chromatography (silica gel, EtOAc/CHCl$_3$ (1:1), Rf = 0.65). Yield: 0.018 g; (38%). Mp: 177–178°C. $^1$H NMR (400 MHz, DMSO-d$_6$, δ): 1.33–1.63 (12H, m, cyclooctane), 1.71–1.82 (2H, m, cyclooctane), 2.95 (3H, s, NCH$_3$), 3.20 (3H, s, NCH$_3$), 3.69 (1H, br.s, CH of cyclooctane), 3.95 (2H, t, J=5.0 Hz, SO$_2$CH$_2$), 4.46 (2H, t, J=5.2 Hz, CH$_2$O), 6.58 (1H, d, J=8.2 Hz, NHCH(CH$_2$)$_2$), 6.99 (1H, t, J=7.3 Hz, CH (C4) of phenyl), 7.26 (2H, t, J=7.8 Hz, CH (C3 and C5) of phenyl), 7.42 (2H, d, J=6.6 Hz, CH (C2 and C6) of phenyl), 8.27 (1H, s, NCHN), 9.66 (1H, s, C(O)NH). $^{13}$C NMR (100 MHz, DMSO-d$_6$, δ): 22.83 (cyclooctane), 25.03 (cyclooctane), 26.54 (cyclooctane), 32.31 (cyclooctane), 35.46 (NCH$_3$), 41.20 (NCH$_3$), 55.27 (CH of cyclooctane, d, J=11.3 Hz), 56.14 (SO$_2$CH$_2$, d, J=2.4 Hz), 57.50 (CH$_2$O), 115.03 (C1, dd, $^1$J ($^{19}$F – $^{13}$C)=13.2 Hz, $^2$J ($^{19}$F – $^{13}$C)=5.9 Hz), 118.48 (C4 of phenyl), 122.64 (C3 and C5 of phenyl), 126.57 (C4, dd, $^1$J ($^{19}$F – $^{13}$C)=18.2 Hz, $^2$J ($^{19}$F – $^{13}$C)=14.1 Hz), 128.62 (C2 and C6 of phenyl), 134.55 (C3, d, J ($^{19}$F – $^{13}$C)=14.5 Hz), 137.09 (C6, ddd, $^1$J ($^{19}$F – $^{13}$C)=241.5 Hz, $^2$J ($^{19}$F – $^{13}$C)=17.6 Hz, $^3$J ($^{19}$F – $^{13}$C)=5.0 Hz), 138.68 (C1 of phenyl), 144.29 (C2, dd, $^1$J ($^{19}$F – $^{13}$C)=253.2 Hz, $^2$J ($^{19}$F – $^{13}$C)=21.3 Hz), 145.66 (C5, ddd, $^1$J ($^{19}$F – $^{13}$C)=250.5 Hz, $^2$J ($^{19}$F – $^{13}$C)=17.2 Hz, $^3$J ($^{19}$F – $^{13}$C)=4.0 Hz), 152.75 (OC(O) NH), 160.76 (NCHN). $^{19}$F NMR (376 MHz, DMSO-d$_6$, δ): –124.61 (1F, s), –134.25 (1F, dd, $^1$J=27.4 Hz, $^2$J=12.3 Hz), 150.25 (1F, dd, $^1$J=27.7 Hz, $^2$J=6.7 Hz). HRMS for C$_{26}$H$_{33}$F$_3$N$_4$O$_6$S$_2$ [(M-H)$^-$]: calc. 617.1721, found 617.1723.

2-((2-(Cyclooctylamino)–3,5,6-trifluoro-4-sulfamoylphenyl)sulfonyl)ethyl phenylcarbamate (**30x**). 2-((2-(Cyclooctylamino)–4-(N-((dimethylamino)methylene)sulfamoyl)–3,5,6-trifluorophenyl) sulfonyl) ethyl phenylcarbamate (**29x**) (0.015 g; 0.024; 1 eq.) and three drops of HCl (conc.) were dissolved in MeOH (3 ml) and refluxed at boiling point for 29 hr. The solvent was evaporated under reduced pressure and the product was purified by column chromatography (silica gel, EtOAc/CHCl$_3$ (1:1), Rf = 0.88). Yield: 0.004 g; (29%). Mp: 174–175°C. $^1$H NMR (400 MHz, DMSO-d$_6$, δ): 1.36–1.63 (12H, m, cyclooctane), 1.75–1.85 (2H, m, cyclooctane), 3.72 (1H, br.s, CH of cyclooctane), 3.97 (2H, t, J=5.3 Hz, SO$_2$CH$_2$), 4.46 (2H, t, J=5.4 Hz, CH$_2$O), 6.62 (1H, d, J=8.3 Hz, NHCH(CH$_2$)$_2$), 6.99 (1H, t, J=7.3 Hz, C4 of phenyl), 7.26 (2H, t, J=7.8 Hz, C3 and C5 of phenyl), 7.43 (2H, d, J=8.0 Hz, C2 and C6 of phenyl), 8.32 (2H, s, SO$_2$NH$_2$), 9.67 (1H, s, OC(O)NH). $^{13}$C NMR (100 MHz, DMSO-d$_6$, δ): 22.74 (cyclooctane), 24.89 (cyclooctane), 26.66 (cyclooctane), 32.15 (cyclooctane), 55.39 (CH of cyclooctane, d, J=11.2 Hz), 56.14 (SO$_2$CH$_2$, d, J=2.8 Hz), 57.45 (CH$_2$O), 114.93 (C1, d, J ($^{19}$F – $^{13}$C)=12.6 Hz), 118.45 (C4 of phenyl), 122.64 (C3 and C5 of phenyl), 127.61 (C4, dd, $^1$J ($^{19}$F – $^{13}$C)=18.6 Hz, $^2$J ($^{19}$F – $^{13}$C)=14.2 Hz), 128.65 (C2 and C6 of phenyl), 134.66 (C3, dd, $^1$J ($^{19}$F – $^{13}$C)=14.2 Hz, $^2$J ($^{19}$F – $^{13}$C)=2.7 Hz), 138.68 (C1 of phenyl), 144.10 (C2, d, J ($^{19}$F – $^{13}$C)=18.2 Hz), 145.59 (C5, ddd, $^1$J ($^{19}$F – $^{13}$C)=250.0 Hz, $^2$J ($^{19}$F – $^{13}$C)=19.3 Hz, $^3$J ($^{19}$F – $^{13}$C)=3.0 Hz), 152.75 (OC(O)NH). $^{19}$F NMR (376 MHz, DMSO-d$_6$, δ): –124.89 (1F, s), –134.36 (1F, dd, $^1$J=27.0 Hz, $^2$J=12.2 Hz), –150.47 (1F, dd, $^1$J=27.0 Hz, $^2$J=6.4 Hz). HRMS for C$_{23}$H$_{28}$F$_3$N$_3$O$_6$S$_2$ [(M+H)$^+$]: calc. 564.1444, found 564.1444.

N-(2-((2-(cyclooctylamino)–3,5,6-trifluoro-4-sulfamoylphenyl)sulfonyl)ethyl)acetamide (**31x**) was prepared according to the known procedure in the literature (*Petrosiute et al., 2024*). N-(2-((2,3,5,6-tetrafluoro-4-sulfamoylphenyl)sulfonyl)ethyl)acetamide (**16x**) (1.39 g; 3.67 mmol; 1 eq.) and cyclooctylamine (1.01 ml; 7.35 mmol; 2 eq.) were dissolved in DMSO (3 ml) and left stirring at room temperature overnight. After full starting material conversion reaction mixture was washed with brine (10 ml) and extracted with EtOAc (3x15 ml). The organic phase was dried using anhydrous Na$_2$SO$_4$ and evaporated under reduced pressure. The product was purified by column chromatography (silica gel, EtOAc Rf = 0.53). Yield 0.900 g; (50%). Mp: 161–162°C (close to the value in the literature [*Petrosiute et al., 2024*], mp: 162–163°C).

## Acknowledgements

This research was funded by the grant from Research Council of Lithuania No S-MIP-22–35. Access to the EMBL beamline P13 at PETRA III (DESY) and has been supported by iNEXT-Discovery, project number 871037, funded by the Horizon 2020 program of the European Commission. This research was also supported within the framework of the European Union's Recovery and Resilience Mechanism project No.5.2.1.1.i.0/2/24/I/CFLA/001 "Consolidation of the Latvian Institute of Organic Synthesis and the Latvian Biomedical Research and Study Centre".

## Additional information

### Competing interests

Aivaras Vaškevičius, Denis Baronas, Asta Zubrienė, Virginija Dudutienė, Daumantas Matulis: has a patent application on CA inhibitors pending. The other authors declare that no competing interests exist.

## Funding

| Funder | Grant reference number | Author |
|---|---|---|
| Research Council of Lithuania | S-MIP-22-35 | Aivaras Vaškevičius<br>Denis Baronas<br>Agnė Kvietkauskaitė<br>Marius Gedgaudas<br>Aurelija Mickevičiūtė<br>Vaida Juozapaitienė<br>Jurgita Matulienė<br>Asta Zubrienė<br>Virginija Dudutienė<br>Daumantas Matulis |
| EU Recovery and Resilience Mechanim | 5.2.1.1.i.0/2/24/I/CFLA/001 | Janis Leitans<br>Kaspars Tars |

The funders had no role in study design, data collection and interpretation, or the decision to submit the work for publication.

## Author contributions

Aivaras Vaškevičius, Conceptualization, Investigation, Methodology, Writing – original draft; Denis Baronas, Data curation, Investigation; Janis Leitans, Agnė Kvietkauskaitė, Audronė Rukšėnaitė, Elena Manakova, Zigmantas Toleikis, Algirdas Kaupinis, Andris Kazaks, Marius Gedgaudas, Aurelija Mickevičiūtė, Vaida Juozapaitienė, Kristaps Jaudzems, Mindaugas Valius, Investigation; Helgi B Schiöth, Kaspars Tars, Franz-Josef Meyer-Almes, Methodology, Writing – original draft; Saulius Gražulis, Jurgita Matulienė, Methodology; Asta Zubrienė, Virginija Dudutienė, Investigation, Methodology, Writing – original draft; Daumantas Matulis, Conceptualization, Resources, Data curation, Formal analysis, Supervision, Funding acquisition, Validation, Visualization, Methodology, Writing – original draft, Project administration, Writing – review and editing

## Author ORCIDs

Aivaras Vaškevičius ⓘ https://orcid.org/0009-0006-8202-9748
Denis Baronas ⓘ https://orcid.org/0000-0003-1878-0837
Franz-Josef Meyer-Almes ⓘ https://orcid.org/0000-0002-1001-3249
Daumantas Matulis ⓘ https://orcid.org/0000-0002-6178-6276

Reviewer #1 (Public review): https://doi.org/10.7554/eLife.101401.3.sa1
Reviewer #2 (Public review): https://doi.org/10.7554/eLife.101401.3.sa2
Reviewer #3 (Public review): https://doi.org/10.7554/eLife.101401.3.sa3
Author response https://doi.org/10.7554/eLife.101401.3.sa4

# Additional files

## Supplementary files
- MDAR checklist

## Data availability

Diffraction data have been deposited in PDB under the accession codes 8OO8, 8S4F, and 9FLF. Compound binding affinity and other thermodynamic data will be entered and freely available for download from the repository https://plbd.org (Protein-Ligand Binding Database). The database information has been published in (*Lingė et al., 2023*).

The following datasets were generated:

| Author(s) | Year | Dataset title | Dataset URL | Database and Identifier |
|---|---|---|---|---|
| Leitans J, Tars K | 2024 | Three-Dimensional Structure of Human Carbonic Anhydrase II in Complex with a Covalent Inhibitor | https://www.rcsb.org/structure/8OO8 | RCSB Protein Data Bank, 8OO8 |
| Leitans J, Tars K | 2024 | Three-Dimensional Structure of Human Carbonic Anhydrase I in Complex with a Covalent Inhibitor | https://www.rcsb.org/structure/8S4F | RCSB Protein Data Bank, 8S4F |
| Leitans J, Tars K | 2024 | Three-Dimensional Structure of Human Carbonic Anhydrase IX in Complex with a Covalent Inhibitor | https://www.rcsb.org/structure/9FLF | RCSB Protein Data Bank, 9FLF |

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
