## [Editor Report · eLife Assessment]

This paper reports the synthesis of covalent inhibitors bearing a unique fragment as a protected covalent warhead for irreversible binding to histidine in carbonic anhydrase (CA) enzymes. These findings are **important** due to the broad utility of the approach for covalent drug discovery applications and could have long-term impacts on related covalent targeting approaches. The data **convincingly** support the main conclusions of the paper.

---

## [Referee Report · Reviewer #1 (Public review)]

Summary:

This paper describes the covalent interactions of small molecule inhibitors of carbonic anhydrase IX, utilizing a pre-cursor molecule capable of undergoing beta-elimination to form the vinyl sulfone and covalent warhead.

Strengths:

The use of a novel covalent pre-cursor molecule that undergoes beta-elimination to form the vinyl sulfone in situ. Sufficient structure-activity relationships across a number of leaving groups, as well as binding moieties that impact binding and dissociation constants.

Weaknesses:

No major weaknesses noted. Suggested corrections were addressed.

---

## [Referee Report · Reviewer #2 (Public review)]

Summary:

The authors utilized a "ligand-first" targeted covalent inhibition approach to design potent inhibitors of carbonic anhydrase IX (CAIX) based on a known non-covalent primary sulfonamide scaffold. The novelty of their approach lies in their use of a protected pre-vinylsulfone as a precursor to the common vinylsulfone covalent warhead to target a nonstandard His residue in the active site of CAIX. In addition to biochemical assessment of their inhibitors, they showed that their compounds compete with a known probe on the surface of HeLa cells.

Strengths:

The authors use a protected warhead for what would typically be considered an "especially hot" or even "undevelopable" vinylsulfone electrophile. This would be the first report of doing so making it a novel targeted covalent inhibition approach specifically with vinylsulfones.

The authors used a number of orthogonal biochemical and biophysical methods including intact MS, 2D NMR, x-ray crystallography, and an enzymatic stopped-flow setup to confirm the covalency of their compounds and even demonstrate that this novel pre-vinylsulfone is activated in the presence of CAIX. In addition, they included a number of compelling analogs of their inhibitors as negative controls that address hypotheses specific to the mechanism of activation and inhibition.

The authors employed an assay that allows them to assess target engagement of their compounds with the target on the surface of cells and a fluorescent probe which is generally a critical tool to be used in tandem with phenotypic cellular assays.

Weaknesses:

This reviewer does not find any major weaknesses beyond those noted in the first round of review.

I understand that some of the previously suggested experiments are cumbersome and I look forward to seeing this manuscript published as well as follow-up on this work in the future.

---

## [Referee Report · Reviewer #3 (Public review)]

Summary:

Targeted covalent inhibition of therapeutically relevant proteins is an attractive approach in drug development. This manuscript now reports a series of covalent inhibitors for human carbonic anhydrase (CA) isozymes (CAI, CAII, and CAIX, CAXIII) for irreversible binding to a critical histidine amino acid in the active site pocket. To support their findings, they included co-crystal structures of CAI, CAII, and CAIX in the presence of three such inhibitors. Mass spectrometry and enzymatic recovery assays validate these findings, and the results and cellular activity data are convincing.

Strengths:

The authors designed a series of covalent inhibitors and carefully selected non-covalent counterparts to make their findings about the selectivity of covalent inhibitors for CA isozymes quite convincing. The supportive X-ray crystallography and MS data are significant strengths. Their approach of targeted binding of the covalent inhibitors to histidine in CA isozyme may have broad utility for developing covalent inhibitors.

Weaknesses:

This reviewer did not find any significant weaknesses. The authors have incorporated most of my suggestions from the first round of review.

---

## [Author Response]

The following is the authors’ response to the original reviews.

**Public Reviews:**

**Reviewer #1 (Public review):**
Summary:This paper describes the covalent interactions of small molecule inhibitors of carbonic anhydrase IX, utilizing a pre-cursor molecule capable of undergoing beta-elimination to form the vinyl sulfone and covalent warhead.Strengths:The use of a novel covalent pre-cursor molecule that undergoes beta-elimination to form the vinyl sulfone in situ. Sufficient structure-activity relationships across a number of leaving groups, as well as binding moieties that impact binding and dissociation constants.Overall, the paper is clearly written and provides sufficient data to support the hypothesis and observations. The findings and outcomes are significant for covalent drug discovery applications and could have long-term impacts on related covalent targeting approaches.Weaknesses:No major weaknesses were noted by this reviewer.
**Reviewer #2 (Public review):**
Summary:The authors utilized a "ligand-first" targeted covalent inhibition approach to design potent inhibitors of carbonic anhydrase IX (CAIX) based on a known non-covalent primary sulfonamide scaffold. The novelty of their approach lies in their use of a protected pre(pro?)-vinylsulfone as a precursor to the common vinylsulfone covalent warhead to target a nonstandard His residue in the active site of CAIX. In addition to a biochemical assessment of their inhibitors, they showed that their compounds compete with a known probe on the surface of HeLa cells.Strengths:The authors use a protected warhead for what would typically be considered an "especially hot" or even "undevelopable" vinylsulfone electrophile. This would be the first report of doing so making it a novel targeted covalent inhibition approach specifically with vinylsulfones.The authors used a number of orthogonal biochemical and biophysical methods including intact MS, 2D NMR, x-ray crystallography, and an enzymatic stopped-flow setup to confirm the covalency of their compounds and even demonstrate that this novel pre-vinylsulfone is activated in the presence of CAIX. In addition, they included a number of compelling analogs of their inhibitors as negative controls that address hypotheses specific to the mechanism of activation and inhibition.The authors employed an assay that allows them to assess target engagement of their compounds with the target on the surface of cells and a fluorescent probe which is generally a critical tool to be used in tandem with phenotypic cellular assays.Weaknesses:While the authors show that the pre-vinyl moiety is shown biochemically to be transformed into the vinylsulfone, they do not show what the fate of this -SO_2_CH_2_CH_2_OCOR group is in a cellular context. Does the pre-vinylsulfone in fact need to be in the active site of CAIX on the surface of the cell to be activated or is the vinylsulfone revealed prior to target engagement?I appreciate the authors acknowledging the limitations of using an assay such as thermal shift to derive an apparent binding affinity, however, it is not entirely convincing and leaves a gap in our understanding of what is happening biochemically with these inhibitors, especially given the two-step inhibitory mechanism. It is very difficult to properly understand the activity of these inhibitors without a more comprehensive evaluation of kinact and Ki parameters. This can then bring into question how selective these compounds actually are for CAIX over other carbonic anhydrases.The authors did not provide any cellular data beyond target engagement with a previously characterized competitive fluorescent probe. It would be critical to know the cytotoxicity profile of these compounds or even how they affect the biology of interest regarding CAIX activity if the intention is to use these compounds in the future as chemical probes to assess CAIX activity in the context of tumor metastasis.
**Reviewer #3 (Public review):**
Summary:Targeted covalent inhibition of therapeutically relevant proteins is an attractive approach in drug development. This manuscript now reports a series of covalent inhibitors for human carbonic anhydrase (CA) isozymes (CAI, CAII, and CAIX, CAXIII) for irreversible binding to a critical histidine amino acid in the active site pocket. To support their findings, they included co-crystal structures of CAI, CAII, and CAIX in the presence of three such inhibitors. Mass spectrometry and enzymatic recovery assays validate these findings, and the results and cellular activity data are convincing.Strengths:The authors designed a series of covalent inhibitors and carefully selected non-covalent counterparts to make their findings about the selectivity of covalent inhibitors for CA isozymes quite convincing. The supportive X-ray crystallography and MS data are significant strengths. Their approach of targeted binding of the covalent inhibitors to histidine in CA isozyme may have broad utility for developing covalent inhibitors.Weaknesses:This reviewer did not find any significant weaknesses. However, I suggest several points in the recommendation for the authors' section for authors to consider.
**Recommendations for the authors:**

**Reviewing Editor Comments:**
The reviewers have made excellent suggestions. We believe a revised version addressing those points can improve the assessment and quality of your work.
**Reviewer #1 (Recommendations for the authors):**
(1) The beta-elimination process is referred to as a "rearrangement" in both the text and the Figure 2 legend. Based on the proposed mechanism the authors provided, it is a simple beta-elimination and conjugate addition mechanism, and is not a rearrangement mechanism. This change should be reflected in the text and Figure 2 legend.

We have made the requested change from rearrangement to elimination reaction.

(2) From a structure-based design perspective, it is not obvious why only large cyclo-alkyl groups were used to target the lipophilic pocket, with the exception of the phenyl carbamates. Perhaps this is background literature on CAIX that describes this? It seems like this is a flexible functional moiety that could be used to impact drug properties. Why were other lipophilic and especially more aromatic or heteroaromatic moieties not studied?

The structure-affinity relationship of the lipophilic ring versus other moieties has been studied and reported previously in manuscripts: Dudutiene 2014, Zubriene 2017, Linkuviene 2018, chapter 16 by Zubriene (https://doi.org/10.1007/978-3-030-12780-0_16). The lipophilic ring served better than a flexible tail or an aromatic ring.

(3) The color-coded "correlation map" in Figure 8 is difficult to follow. Perhaps a standard SAR table with selectivity and affinity values would be easier to read and follow.

We are trying to promote “correlation maps” because in our opinion they are easier to follow than tables.

(4) Although there is a statement for this in line 254 of the SI, the compound numbering in the SI, vs. the numbering used in the manuscript is confusing. The standard format for these is to consecutively number all compounds and have identical compound numbers in both the SI and manuscript. The synthetic intermediates included in the SI can be identified by IUPAC names.

An additional numbering system had to be made because the synthesis was described in the supplementary materials. We would prefer to leave the numbering as in the current manuscript. There are quite a few intermediate compounds that we assigned intermediate numbers such as 20x in order to make it simpler to distinguish intermediate synthesis compounds from compounds that were studied for binding affinity.

(5) Ranges of isolated yields for the synthetic steps in SI schemes SI, S2, and S3 need to be included.

We have remade the SI schemes S1, S2, and S3 to include the yields of each compound.

(6) Presumably, the AcOH/H2O2 reaction forms the sulfones and not sulfoxides when heat is used. In the SI, the structures of 9x and 10x are shown to be sulfoxides and not sulfones. Initially, this is thought to be a simple structural mistake, however, this is concerning, since the HRMS data (for compound 9x) reported is for the sulfoxide (HRMS for C8H7F4NO4S2 [(M+H)+]: calc. 321.9825, found 321.9824. 482) and not the sulfone? In the synthesis scheme S1, condition "C" is used for both the sulfoxide and sulfone synthesis (i.e. 3ax to 9x vs. 12x to 13x). It appears the sulfoxide is prepared using a room temperature procedure, vs. the sulfone requiring 75 degrees centigrade heat. These two similar conditions need to be designated as different synthetic steps in the schemes with the specific conditions noted since the products formed are different.

We have made requested corrections/adjustments and added separate reaction conditions for sulfoxide synthesis in SI scheme S1.

**Reviewer #2 (Recommendations for the authors):**
I appreciate that it's difficult to determine parameters such as kinact or Ki of such potent inhibitors and ones that work by a two-step mechanism. I might suggest characterizing the steps separately to determine the detailed parameters. Maybe something like NMR for the for the activation step and SPR for the kinact and Ki of the unmasked vinylsulfone?

We agree that such information would be helpful. However, it requires significant effort and equipment and will be performed in a separate study.

I always advocate for at least a global proteomics analysis using a pulldown probe to get an idea of the specificity profile, especially for the so-far untried and untested pre-vinylsulfone moiety.

We fully agree that the pull-down assay is a good idea. However, this major task will be performed in a separate study.

This might be picky but wouldn't this be considered a pro-vinylsulfone rather than pre-vinylsulfone? Just as the term "prodrug" is used?

We agree that both the pre-vinylsulfone and pro-vinylsulfone are suitable names. However, in pharmacology, the prodrug is common, but in organic synthesis, the precursor is commonly used. Therefore, we prefer to keep the pre-vinylsulfone.

I would also be curious to know what species is responsible for activating the compound to the vinylsulfone. Maybe make some key point mutations of nearby basic residues?

The His64 formed the covalent bond, thus His64 was the likely activating base. Preparing a mutation could be a good path for future studies.

**Reviewer #3 (Recommendations for the authors):**
(1) The authors presented only a close-up view of the active site with a 2Fo-Fc map mesh in three panels of Figure 4. For readers unfamiliar with the carbonic anhydrase field, adding a complete illustration of each protein-inhibitor complex (protein in cartoon mode and ligand in stick) will be helpful. Also, an image of the 180º rotation of the close-up view presented in each panel should be added. Depicting h-bonds between critical residues (Asn62, Gln 92, etc.) with dashed lines and marking the distances will be helpful for readers.

We have prepared a requested picture for CAIX. Panels on the left show entire protein molecule view of the bound ligands to each isozyme and there are two close-up views for each structure rotated 180 degrees.

(2) Line 198 should be revised to refer to the correct complexes. 20, 21, and 23 should be 21, 20, 23.

We appreciate that the reviewer noticed this error. We corrected the mistake.

(3) Omit electron density maps around each ligand in Figure 4 should be included for compounds 20, 21, and 23, perhaps as a supplementary figure.

Detailed electron density map information is provided in the mtz files that have been submitted to the PDB. We think the omit maps are not necessary in the supplementary materials.

(4) The cyclooctyl group is stabilized by hydrophobic active site residues, L131, A135, L141, and L198. However, only L131 is shown in Figure 4. All residues that stabilize the ligands should be shown.

For clarity purposes of the figure, we have omitted some of the residues that make contact with the ligand molecule. We think that the structure provided to the PDB could be analyzed in detail to see all contacts between the ligand and protein molecule.

(5) The supplementary table S1 lacks the crystallographic data on the CAIX-23 complex.

We have added a new version of the supplementary materials that contains the crystallographic data on the CAIX-23 complex.

(6) A minor peak (30213 Da) with a 638 Dalton shift compared to the unmodified enzyme is for Figure 5A, not Figure 5B, as mentioned in line 235. This sentence in line 235 should be corrected.

We corrected this mistake.

(7) As the authors stated in the text, a minor peak (30213 Da) represents a potential second binding site. Can they revisit their electron density maps and show any residual density if it is present around a second histidine residue? The MS data in Figure S17C indicates the presence of additional sites for compound 12. Thus, additional electron density around the secondary and tertiary sites is possible.

CAII contains His3 and His4 that are at the N-end of the protein and not visible in the crystal structure. The NMR data indicate that the additional modification may occur at one of these His residues.

(8) MS data were presented for compounds 12 and 22 in Figure 5A, B, but the co-crystal structures were generated with compounds 21, 20, and 23. Why was no MS data included for compounds 20, 21, and 23? Would these compounds show the presence of a secondary binding site? Can authors include the MS data?

In the main body of the manuscript in Figure 5A we only present MS data on CAXIII with compound 12. It is only an example that confirms covalent interaction. In the supplementary we have MS data for compound 12 with all carbonic anhydrase isozymes and compound 20 with almost all (except CAVI) CA isozymes. There are also MS data provided with numerous compounds (3, 9, 13, and other) and CA isozymes that serve as a control or confirmation of covalent bond formation.

(9) The coordination between the zinc ion and NH of the ligand is mentioned in the enzyme schematic in Figure 3. Can the distances and coordination with Zinc be illustrated in ligand-bound structures in Figure 4?

We considered and decided that picture which shows the numerous distances between ligand atoms and protein residues would be difficult to follow. The structures provided to the PDB could be analyzed for every aspect of the complex structure.

(10) A key difference between covalent (compound 12) and its non-covalent counterpart, compound 5, is the two oxygens attached to sulfur in compound 12. Do protein side chains or water interact with these oxygens? Are these oxygen atoms exposed to solvent? Can authors show the interactions or clarify if there is no interaction?

The two oxygens in the ligand molecule serve several purposes. First, they pull out electrons and diminish the pKa of the sulfonamide, thus making interaction stronger. Second, the oxygen atoms may make contacts, hydrogen bonds with the protein molecule and may also be important for covalent bond formation. Exact energy contributions cannot be determined from the structure directly. Thus, we decided to not yet explore and delve into this area.

(11) Fix the font size of the text in lines 355-356.

The font has been corrected.